# Self-assembled dendrimer polyamide nanofilms with enhanced effective pore area for ion separation

Bingbing Yuan [1] ✉, Yuhang Zhang[1], Pengfei Qi[2], Dongxiao Yang[1], Ping Hu[1], Siheng Zhao[1,3], Kaili Zhang[1], Xiaozhuan Zhang[1], Meng You[1], Jiabao Cui[1], Juhui Jiang[1], Xiangdong Lou[1] & Q. Jason Niu [3] ✉

Membrane technology using well-defined pore structure can achieve high ion purity and recovery. However, fine-tuning the inner pore structure of the separation nanofilm to be uniform and enhance the effective pore area is still challenging. Here, we report dendrimers with different peripheral groups that preferentially self-assemble in aqueous-phase amine solution to facilitate the formation of polyamide nanofilms with a well-defined effective pore range and uniform pore structure. The high permeabilities are maintained by forming asymmetric hollow nanostripe nanofilms, and their well-designed ion effective separation pore ranges show an enhancement, rationalized by molecular simulation. The self-assembled dendrimer polyamide membrane provides $Cl^-/SO_4^{2-}$ selectivity more than 17 times that of its pristine polyamide counterparts, increasing from 167.9 to 2883.0. Furthermore, the designed membranes achieve higher Li purity and Li recovery compared to current state-of-the-art membranes. Such an approach provides a scalable strategy to fine-tune subnanometre structures in ion separation nanofilms.

Ion separation is significant in fields such as zero discharge of industrial wastewater and lithium extraction from salt lakes, but these achievements of processes are highly energy-intensive and complex[1,2]. Membrane-based ion sieving technologies can address these tricky issues because their separation involves no phase change and their good process coupling capability. Polyamide nanofilms formed on ultrafiltration supports via interfacial polymerization (IP) have been confirmed as advanced desalination or separation membranes[3,4]. However, traditional polyamide (PA) chemistry made by trimesoyl chloride (TMC) and piperazine (PIP) via diffusion-polymerization generally tends to form nanopores characterized by multiscale heterogeneity and nonuniformity due to their rapid and random crosslinking reaction, exhibiting limited ion separation precision and permeance[5,6]. In addition, similar to desalination membranes that produce freshwater from seawater using reverse osmosis, ideal ion separation membranes still need to break the trade-off between ion permselectivity and ion/water permeability to achieve high ion purity and ion recovery[7,8].

The implementation of membrane-based separations is based on molecule/ion differences in size, charge or interaction[9]. The acquisition of high separation precision is inseparable from the fine-tuning of nanofilm nanopores or charges[10,11]. More importantly, the uniformity and effective separation range of nanofilm pores should also be considered[12]. The main approaches to tuning nanofilm pores and charge include: (1) the addition of surfactants[5,13], salts[14,15], artificial water channels[16–18], ionic dendrimers[19], or nanofillers[20,21], (2) surface

[1]School of Chemistry and Chemical Engineering, Key Laboratory of Green Chemical Media and Reactions Ministry of Education, Henan International Joint Laboratory of Aquatic Toxicology and Health Protection, Henan Normal University, 453007 Xinxiang, China. [2]State Key Laboratory of Separation Membranes and Membrane Processes, National Center for International Research on Membrane Science and Technology, School of Materials Science and Engineering, Tiangong University, Tianjin 300387, P. R. China. [3]Institute for Advanced Study, Shenzhen University, Nanshan District Shenzhen 518060 Guangdong, China. ✉e-mail: yuanbingbing@htu.edu.cn; qjasonniu@szu.edu.cn

grafting or modification[22], and (3) modifying monomer structure[23]. An improved permeance-selectivity scope was observed for the obtained membranes, but the trade-off curve reported remained to be broken[8]. Taking the Li$^+$ separation from Mg$^{2+}$ as an example, polyethylenimine (PEI)-based PA membranes showed high MgCl$_2$ rejection due to their positively charged surface, whereas a lack of fine-tuning of the nanopore size resulted in high LiCl rejection as well, showing compromised Li$^+$/Mg$^{2+}$ selectivity. Meanwhile, the valuable operational trade-off line elucidated between Li$^+$/Mg$^{2+}$ permselectivity and Li$^+$/water permeability indicated that Li purity and Li recovery need to be balanced to obtain high-efficiency ion separation[8]. Therefore, developing an efficient nanopore range and uniform pore structure within the separating nanofilm for high ion permeselectivity is crucial for Li extraction from salt lakes.

Dendrimers featuring ordered, well-defined subnanoscale pores and precise molecular structures have been regarded as ideal nanomaterials for ion or molecule separation. However, these macromolecules are hardly soluble in water, making it impossible for them to participate in IP to form nanofilms to impart separation. Hence, we speculate that one of the key challenges in maximizing the inner well-defined pore structure advantages of dendrimers is how to orderly assemble them within polyamide nanofilms to enhance ion separation.

Here, we show self-assembled dendrimer (SAD) polyamide nanofilms with nanostripe asymmetric structures that break the current trade-off between Li$^+$/Mg$^{2+}$ permselectivity and Li$^+$/water permeability. The introduction of an asymmetric structure with a nanostripe in the polyamide nanofilm confirms that it enables an increase in permeance. Our well-designed strategy elucidates that the positively charged SADs formed in aqueous-phase amine solution can participate in the formation of the dense PA nanofilm, leading to fine-tuning of the nanopore structure within the nanofilm and facilitating ion separation. One highlight of this study is that these designed dendrimers of different functional groups (aliphatic chained carboxyl, aromatic carboxyl and phenolic hydroxyl) self-assembled into octahedral, cube or spherical nanoparticles. Moreover, all fine-tuned polyamide nanofilms with SADs demonstrate enhanced effective separation pore areas and uniform pore structures, boosting Li$^+$/Mg$^{2+}$ and Cl$^-$/SO$_4^{2-}$ permselectivity. This work indicates that using a self-assembled dendrimer solution as an aqueous-phase reaction system to conduct interfacial synthesis is a particularly important strategy to fine-tune the nanofilm inner pore structure.

## Results

### Formation of SAD aqueous-phase PIP solution

Generally, aromatic polyamide dendrimers are difficult to dissolve in water due to their larger molecular weight and intramolecular hydrogen bonding. Deprotonation of dendrimers can enhance their ion-hydration free energy, making it possible to dissolve in PIP solution. Dendrimers with carboxyl and phenolic hydroxyl groups on their periphery were designed and synthesized (Fig. 1a, Supplementary Figs. 1–3). Nuclear magnetic resonance (NMR) and high-resolution mass spectral (HRMS) confirmed their structure and molecular weight (Supplementary Figs. 4–13). Figure 1b–d show that, the single dendrimer nanoparticle sizes of DA-G4D, BA-G4D and p-HC-G4D were approximately 6.7 nm, 5.6 nm and 6.2 nm, respectively (Supplementary Figs. 14–16 and 25d). Brunauer Emmet Teller N$_2$ adsorption–desorption experiments showed a comparable surface area of the resultant dendrimers (Supplementary Fig. 17).

After deprotonation in PIP solution, the carboxyl- and phenolic hydroxyl-based dendrimers were dissolved and formed a stable solution. The images of the transparent sample bottle containing solution in Fig. 1b–d show electronic photos of the SADs aqueous-phase PIP solution after stable storage for 1 year. More importantly, self-assembled behaviour was found when DA-G4D, BA-G4D and p-HC-G4D dendrimers were dissolved in PIP solution. Due to the

electrostatic interaction difference between PIP solution and carboxyl and phenolic hydroxyl- dendrimers, the DA-G4D SADs exhibited octahedral morphology (Fig. 1b, Supplementary Figs. 18, 19), the BA-G4D SADs showed cubes (Fig. 1c, Supplementary Figs. 20, 21), and the p-HC-G4D SADs were spherical (Fig. 1d, Supplementary Figs. 22, 23). Figure 1e further shows the SEM micrographs of aggregated DA-G4D SADs, presenting a regular octahedral structure. Overall, the diameters of the DA-G4D, BA-G4D and p-HC-G4D SADs from SEM images were 119.8 ± 21.88 nm, 138.57 ± 20 nm and 51.1 ± 3.33 nm, respectively (Supplementary Fig. 24). Closer HRTEM imaging from Fig. 1f demonstrated that the diameter of the nanovoids in these SADs ranged from 0.96 nm to 1.12 nm. Hence, based on the hydrated radius of Mg$^{2+}$ (0.428 nm), Cl$^-$ (0.332 nm), SO$_4^{2-}$ (0.379 nm) and Li$^+$ (0.382 nm), these nanovoids effectively allowed the passage of ions. Both HRTEM micrographs and SEM images confirmed the morphology of the SADs.

To obtain insight into the formation mechanism of SADs, we studied nanoparticle size and charge for different concentrations of PIP and dendrimer using dynamic light scattering (DLS) and zeta potential analyser (Supplementary Figs. 25–27). When the PIP was 0.01 wt.% and the dendrimer was 0.01 wt.%, DA-G4D still exhibited the single particle morphology, whereas BA-G4D and p-HC-G4D showed a self-assembled morphology with a larger size (Supplementary Fig. 25a). We estimated that DA-G4D behaves differently because it has an aliphatic chain carboxyl group, which is more likely to move in aqueous solution rather than anchor by electrostatic interactions, than aromatic carboxyl (BA-G4D) and phenolic hydroxyl groups (p-HC-G4D). In other words, DA-G4D dendrimers with aliphatic chain carboxyl groups need more PIP molecules to generate electrostatic interactions and form large self-assembled dendrimers (SADs). This speculation was supported in Supplementary Fig. 25b; that is, when the PIP concentration was increased up to 0.02 wt.%, 0.03 wt.% and 0.05 wt.%, the sizes of the DA-G4D SADs became large accordingly. For highly concentrated solutions, for example, containing 0.04 wt.% dendrimer and 0.8 wt.% PIP, an SAD morphology 100 ~ 200 nm in size was observed (Supplementary Fig. 25c). Single dendrimer nanoparticles were again observed when a high-concentration solution (0.8 wt.% PIP and 1 wt.% dendrimer) was diluted 20 times using 0.1 wt.% PIP (Supplementary Fig. 25d). The dilute concentrations calculated for PIP and dendrimer are 0.135 wt.% and 0.05 wt.%, respectively. The zeta potentials of these SADs before and after dilution demonstrated that using 0.1 wt.% PIP to dilute the SAD solution can alter the electrostatic interaction between dendrimers and PIP molecules, further resulting in the disaggregation of the SADs formed and finally turning into single dendrimer nanoparticles (Supplementary Fig. 26). This result indicated that the formation mechanism of SADs was electrostatic self-assembly, and correlated with the PIP and dendrimer concentrations.

The peripheral charge for SADs was further studied (Supplementary Fig. 27). To be specific, we studied the effect of PIP and dendrimer concentrations on the surface potential of the SADs. The DA-G4D, BA-G4D and p-HC-G4D SADs exhibited a negative surface charge under 0.01 wt.% PIP and 0.01 wt.% dendrimer (Supplementary Fig. 27a–c). With increasing of PIP and dendrimer concentrations (0.8 wt.% PIP and 0.04/0.08 wt.% dendrimer), the DA-G4D, BA-G4D and p-HC-G4D SADs possessed a positive surface charge, which indicated that the peripheries of SADs featured aggregated PIP molecules could be involved in IP to form amide bonds and fine-tuned the nanofilm inner structure (Supplementary Fig. 27d–f).

These conditions explored the regularity of SAD formation and the corresponding structural features, and produced the possible best concentration of SAD aqueous-phase PIP solution for IP asymmetric nanofilm fabrication. In fact, there were several criteria for selection in this study: the direct desalination or ion separation performance of the resulting membranes and the zeta potential or morphology structure of SADs. For example, a positive zeta potential value ensured

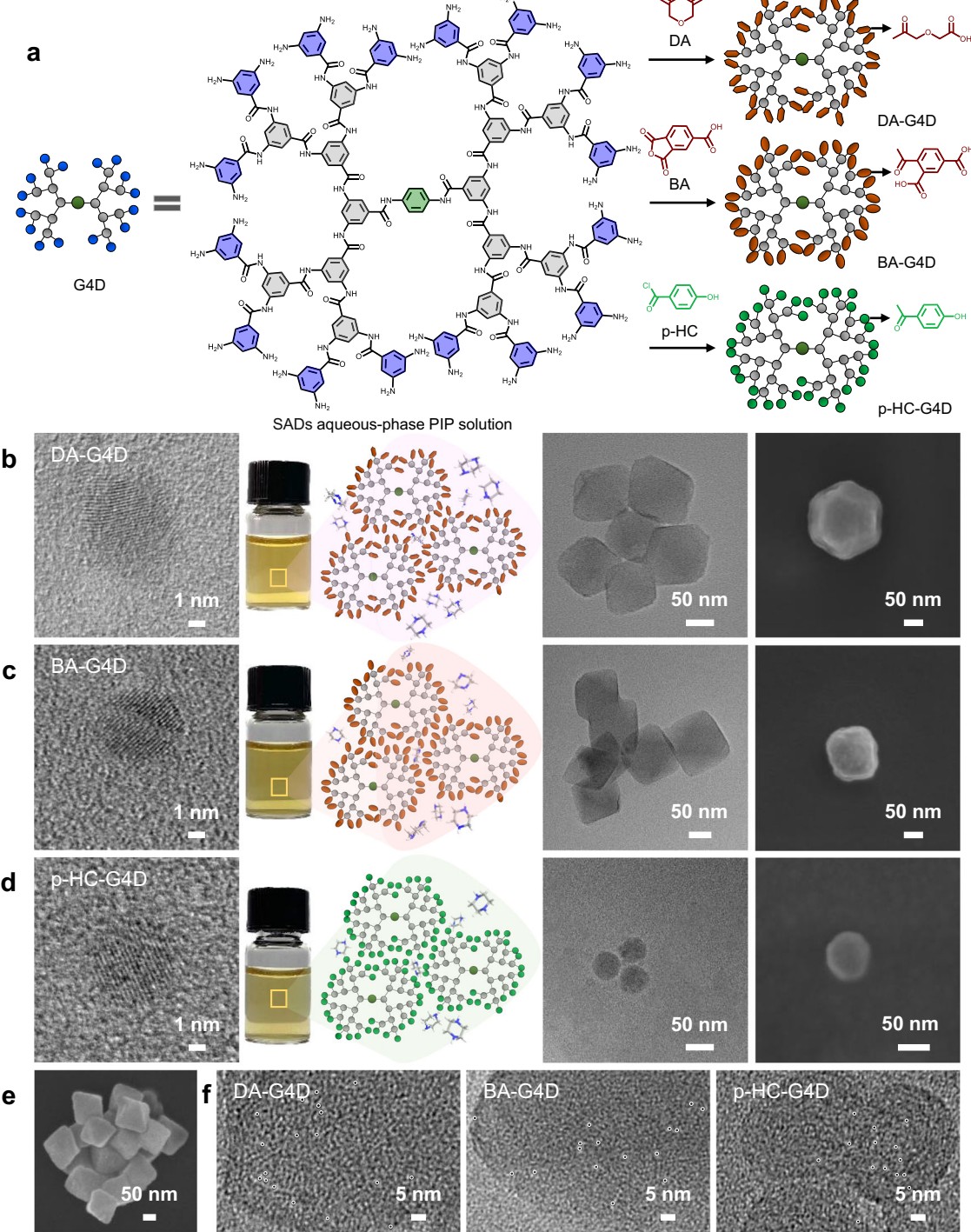

**Fig. 1 | Fabrication and characterizations of self-assembled dendrimer (SAD) aqueous-phase PIP solution. a** Synthetic route to different functional dendrimers (DA-G4D, BA-G4D and p-HC-G4D) using reactive monomers, diglycolic anhydride (DA), 1, 2, 4-benzenetricarboxylic anhydride (BA), and p-hydroxybenzoyl chloride (p-HC). **b–d** Schematic and photographs of SAD aqueous-phase PIP solution, which was characterized by high-resolution transmission electron microscopy (HRTEM) and scanning electron microscopy (SEM), showing self-assembled octahedra (DA-G4D), cubes (BA-G4D) or spheres (p-HC-G4D). Among them, the HRTEM images labelled DA-G4D, BA-G4D and p-HC-G4D represent the morphologies of its single dendrimer. **e** SEM morphology image of the DA-G4D SADs. **f** Closer HRTEM imaging of the DA-G4D, BA-G4D and p-HC-G4D SADs. The white circles in the images represent nanovoids.

that there were sufficient PIP molecules on the periphery of SADs, which can participate in the IP process, leading to the enhancement of interface compatibility and the stability of imbedding. Hence, we rationally speculated that for A-PA/DA-G4D, the best conditions were 0.8 wt.% PIP and 0.04 wt.% DA-G4D. The best conditions for A-PA/BA-G4D were 0.8 wt.% PIP and 0.04 wt.% BA-G4D, while those of A-PA/p-HC-G4D were 0.8 wt.% PIP and 0.08 wt.% BA-G4D.

## Formation and structural mhorphologies of SAD polyamide nanofilms

To better fabricate the polyamide nanofilms with defect-free and high permeation flux, IP was conducted on the dendrimer porous layer formed via a diazotization-coupling reaction of the polysulfone (PSF) support[24]. Specifically, after formation of the dendrimer porous layer on the PSF support, compared with the pristine PSF support, the polar

part of the solid surface energy was increased, while that of the non-polar part was decreased, and its pore size was also decreased[24]. When the aqueous PIP solution was soaked into the PSF support with a dendrimer porous layer, it interacted with the dendrimer porous layer via hydrogen bonding and led to an inhomogeneous PIP solution distribution, reducing the diffusion rate of the PIP molecule[4]. Hence, the diffusion-reaction behaviours of PIP molecules from the aqueous phase to the organic phase conducted on the modified PSF support with a dendrimer porous layer were different from those of the pristine PSF support. It can be rationally speculated that such a diffusion reaction was inconsonant or nonuniform at the nanoscale, where some IP reaction sites were dominated by the dendrimer porous layer, whereas other sites were still affected by the PSF support. This diffusion-driven instability facilitates the formation of a nanostripe structure[4]. Furthermore, the remaining amine groups on the periphery of the dendrimer porous layer after the diazotization-coupling reaction enabled participation in the IP reaction with TMC, forming a polyamide nanofilm with a two-layer nanostructure via stable covalent bonds[24]. Finally, the formed polyamide nanofilms exhibited nanostripe and asymmetric structures, showing a high specific surface area and more permeable channels[4]. With the formation of an SAD aqueous-phase PIP solution, when applied in the IP stage, peripheral PIP molecules of the SADs can react with the TMC during IP process to form amide bonds, leading to the enhancement of their interface compatibility and the stability of imbedding. Then, SAD polyamide nanofilms with nanostripe and asymmetric structures were formed.

The upper surface images from SEM (Fig. 2b–d, Supplementary Figs. 28–30) showed the existence of SAD (DA-G4D, BA-G4D and p-HC-G4D) nanoparticles that were embedded within the surroundings of the nanostripe PA layer, demonstrating good compatibility. This stemmed from the aggregative PIP molecules on the periphery of the SADs that can be involved in the interfacial reaction. In contrast, they were not observed in the pristine A-PA membrane (Fig. 2a, Supplementary Fig. 31). Further inspection demonstrated that the sizes of the SADs in the upper surface of the PA layer were matched with the HRTEM micrographs. TEM images of the bottom to top nanofilms revealed a hollow nanostripe structure among the obtained PA nanofilms (Fig. 2, Supplementary Figs. 32–35). The typical widths of the nanostripe ranged from 100～200 nm, and their inner hollow parts were ~100～150 nm. These nanostripes and their internal hollows increase the permeable area of water, and optimize the water transport channel[4,21,24].

Close observation of the SEM cross-sections showed an asymmetric structure (Fig. 2). The sublayer was a dendrimer porous layer with a thickness of 50–70 nm, while the top layer consisted of a dense PA layer. However, the thickness of the upper layer observed with the incorporation of SADs was clearly more prominent than that of the original A-PA nanofilm (Fig. 2a). This was also confirmed by TEM images of the cross-sectional morphology (Supplementary Figs. 36–39). Both SEM and TEM cross-section images demonstrated that the dense layer thickness of the A-PA nanofilm was ~40 nm, which was lower than those of the A-PA/DA-G4D (50 nm), A-PA/BA-G4D (60 nm), and A-PA/p-HC-G4D (50 nm) nanofilms. We rationally speculated that there are two main reasons. One is because the SAD solution, featuring by aggregated PIP molecules on their peripheries, can be located at the interface of the aqueous and organic phases during the IP process, indirectly increasing the PIP concentration at the interface to facilitate the formation of a thicker polyamide nanofilm. Another is that these SADs with intrinsic nanoscale sizes and aggregated PIP molecules can participate in IP (Supplementary

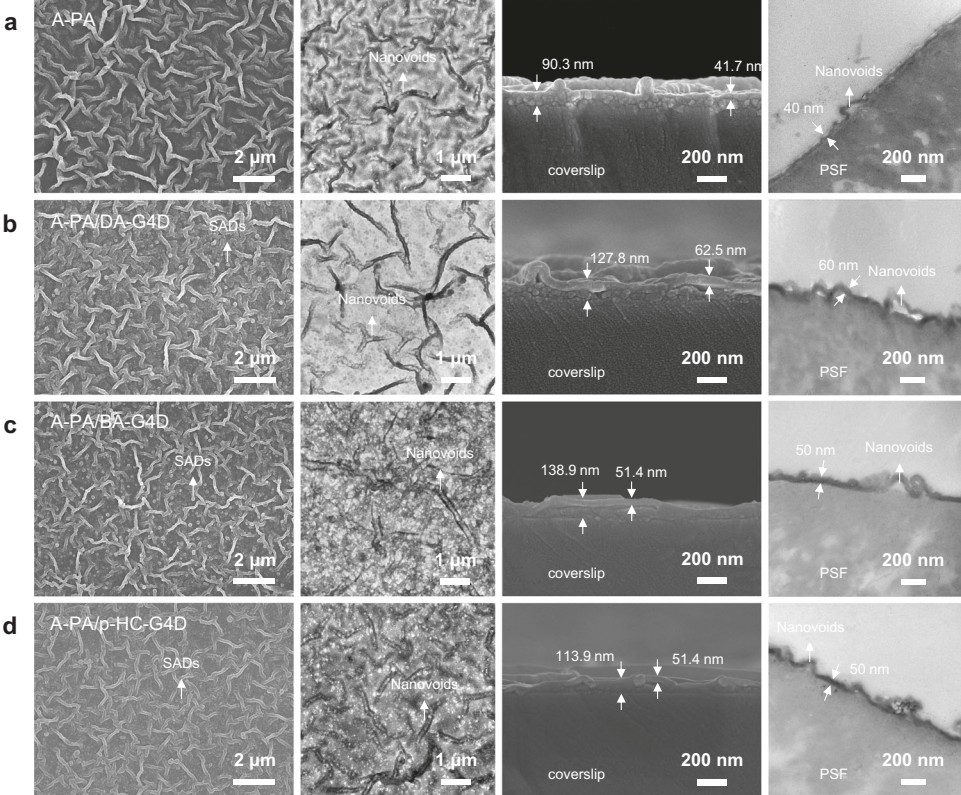

**Fig. 2 | Morphologies and microstructures of resulting nanofilms. Pristine asymmetric polyamide (PA) membrane ((a) A-PA) and the self-assembled dendrimers (SADs) PA membranes (b) A-PA/DA-G4D (c) A-PA/BA-G4D, and (d) A-PA/p-HC-G4D).** From left to right, the upper surfaces of nanofilms characterized by SEM, bottom to top micrographs of nanofilms from TEM, and cross-sectional morphology of nanofilms characterized by SEM supported by coverslip and TEM supported by polysulfone (PSF) support. SEM images of the upper surfaces reveal the incorporation of SADs into the nanofilms.

Figs. 25–27) and be embedded into the polyamide nanofilm, then increase the thickness of the polyamide nanofilm.

Moreover, for cross-section TEM images, the resin used for slicing readily permeated into the dendrimer porous layer, such as by permeating the PSF support with an inner loose structure. The sublayer (dendrimer porous layer) thus cannot be observed in TEM imaging, as in the SEM cross-section images. In addition, nanovoids consisting of nanostripes were clearly observed in the cross-sectional morphology of nanofilms characterized by TEM and the bottom to top micrographs of nanofilms obtained from TEM and SEM, where such morphology also contributed to providing a higher water permeation rate.

Volumetric charge densities calculated from zeta potential values at pH = 7 showed an enhanced negative charge when the SADs were imbedded into the PA layers (Fig. 3a, Supplementary Fig. 40). This is because the intrinsic charge of the carboxylic acid group and phenolic hydroxyl group in the periphery of dendrimers (DA-G4D, BA-G4D and p-HC-G4D) is negative, as confirmed in Supplementary Fig. 27a–c. The

presence of SADs was also demonstrated by results including volumetric density (Fig. 3a), O/N ratio and functional group contents from N1$s$ and O1$s$ of peak fitting (Fig. 3b). The volumetric density was obtained by using the mass from a quartz crystal microbalance (QCM) with dissipation, and the thickness data from cross-sectional SEM images. The calculated volumetric density slightly increased when SAD polyamide nanofilms formed, ranging from $1.23 \pm 0.063$ g cm$^{-3}$ (for A-PA) to $1.30 \pm 0.085$ g cm$^{-3}$ (for A-PA/DA-G4D), $1.31 \pm 0.065$ g cm$^{-3}$ (for A-PA/BA-G4D), and $1.30 \pm 0.084$ g cm$^{-3}$ (for A-PA/p-HC-G4D). The O/N ratios determined by XPS exceeded the pristine TMC−PIP nanofilm (1.51), which was as high as 5.25 for the A-PA/DA-G4D, 5.65 for the A-PA/BA-G4D, and 6.47 for the A-PA/p-HC-G4D, confirming the formation of SAD polyamide nanofilms as well. This can also be verified through the variation in NH−C = O content from N1$s$ (Supplementary Fig. 41).

The pore size distributions can be calculated by the neutral solute transport method (Supplementary Fig. 42). The results showed that narrower mean effective pore sizes ($\mu_p$) were observed within the SADs

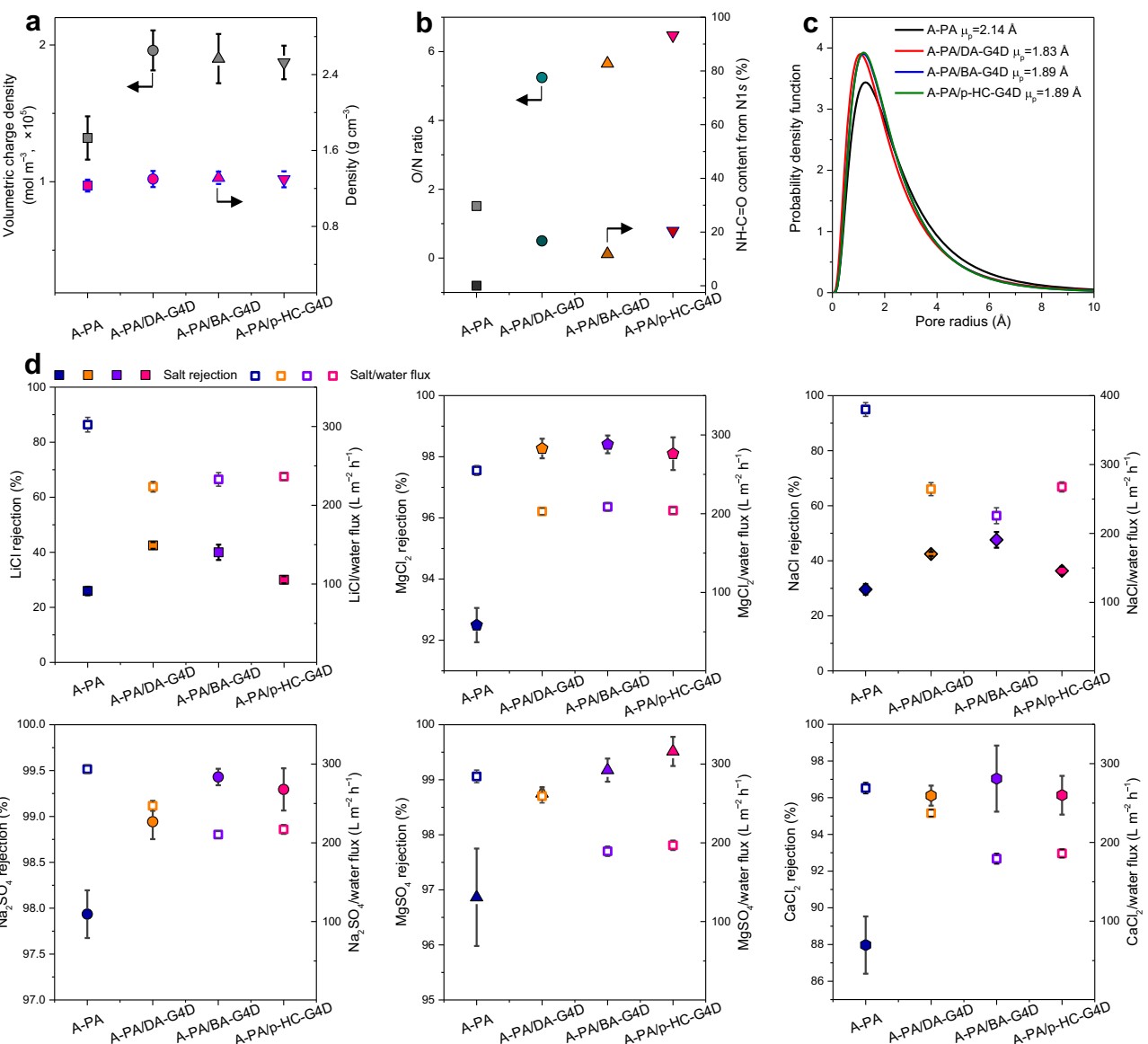

**Fig. 3 | Chemical structure analysis and performance of the pristine asymmetric polyamide (PA) membrane (A-PA) and the SADs PA membranes (A-PA/DA-G4D, A-PA/BA-G4D, and A-PA/p-HC-G4D). a** Volumetric charge density and volumetric density. **b** O/N ratio and NH−C = O content from N1$s$, which was characterized by X-ray photoelectron spectroscopy (XPS). **c** Pore distribution that was calculated from the neutral molecule transport method. **d** Single salt rejection and salt/water flux of the A-PA and SAD polyamide membranes. The error bars represent the reproducible data obtained from at least three independent membrane samples.

PA layer in comparison with the A-PA nanofilm (Fig. 3c). For example, the mean effective pore size of A-PA was 2.14 Å, while those of the SAD PA nanofilms were decreased to 1.83 Å (for A-PA/DA-G4D), 1.89 Å (for A-PA/BA-G4D) and 1.89 Å (for A-PA/p-HC-G4D).

These results demonstrate that using an SAD PIP solution as an aqueous-phase reactive system can enhance the negative charge and narrow the mean effective pore size of the resultant nanofilms. In addition, the formed membranes basically showed equivalent wettability (Supplementary Fig. 43). Hence, when the difference of the resultant membranes in terms of water/salt permeance flux was studied and discussed, the wettability would no longer be taken into account as a factor to influence flux. We will consider the influence of thickness and surface morphology on the water/salt permeance flux.

## Desalination performance

Pore size and charge are essential for ion separation because they impact the rejection of $Mg^{2+}$ or $SO_4^{2-}$ and the permeation of $Li^+$ or $Cl^-$[10-12]. Moreover, the defect-free distribution of the hollow nanostripe structure and nanofilm thickness are essential to increase the water permeation performance since they determine the pathways for water flow across the active nanofilm and the corresponding transmembrane resistance. Here, SAD PA membranes with nanostripe asymmetric structures were first tested under single salt conditions, a cross-flow pressure of 10 bar, and six representative salts (Fig. 3d).

Specifically, after exploring the conditions used for the fabrication of the control polyamide membrane, we fixed the related PIP and TMC concentrations to 0.8 wt.% and 0.1 w/v%, respectively. Then, different dendrimer concentrations ranging from 0.02 wt.%, 0.04 wt.%, 0.06 wt.%, 0.08 wt.% and 0.1 wt.% were added to 0.8 wt.% PIP to generate an SAD reactive solution and then reacted with TMC to form SAD polyamide membranes. Through conducting single salt separation performance tests, analysis and comparison, we determined that the optimized conditions to fabricate the SAD polyamide nanofilms were as follows. For A-PA/DA-G4D, the optimized conditions were 0.8 wt.% PIP and 0.04 wt.% DA-G4D. The optimized conditions for A-PA/BA-G4D were 0.8 wt.% PIP and 0.04 wt.% BA-G4D, while that of A-PA/p-HC-G4D were 0.8 wt.% PIP and 0.08 wt.% BA-G4D. Compared with the control A-PA membrane, optimized asymmetric PA membranes comprising SADs (DA-G4D, BA-G4D and p-HC-G4D) showed a marked enhancement in salt rejection and a coordinate decrease in salt/water flux. Specifically, the $MgCl_2$ rejection increased from 92.5% for the control A-PA membrane to >98% or even 99% for the SADs PA membranes. Meanwhile, the relevant $MgCl_2$/water flux decreased from 275 to 200 L m$^{-2}$ h$^{-1}$ due to the increased thickness of the dense polyamide layer, still exhibiting a higher water flux. Moreover, LiCl maintained a lower rejection, which was in the range of 30 - 42.5%. Correspondingly, the water fluxes for LiCl showed a reduction from 302.32 to 223.53 L m$^{-2}$ h$^{-1}$.

On the whole, a notable difference in salt rejection between $MgCl_2$ and LiCl was achieved, which was better than most positively charged PA membranes. This is because, high $MgCl_2$ rejection rates are often accompanied by high LiCl rejection rates for PA membranes with a positive charge[22,25]. Hence, our SADs PA membranes displayed a better combination of salt/water flux and LiCl/$MgCl_2$ difference compared to the literature reported PEI-based[22,25], marketed (NFX, DK, DL, NF 90 and NF 270)[8,26], surface modified[27] and other doping state-of-the-art membranes[28]. On the other hand, under the optimized A-PA membranes incorporated SADs, the $Na_2SO_4$ rejection rate was >99% (Fig. 3d). The NaCl rejection varied from 30 ~ 40% (Fig. 3d). For salt/water flux, the corresponding NaCl/water flux was in the range of 230 to 280 L m$^{-2}$ h$^{-1}$, while those of the $Na_2SO_4$/water fluxes were as high as 210 to 250 L m$^{-2}$ h$^{-1}$, maintaining a comparable salt/water permeation flux.

## Ion separation performances and permeabilities

Separation of $Li^+$/$Mg^{2+}$ and $Cl^-$/$SO_4^{2-}$ is considered as a significant approach for achieving circularity in resource management, such as

salt lake lithium extraction and industrial wastewater minimal or zero-liquid discharge strategies[2]. A series of mixed ion experiments were conducted to explore the separation selectivity in terms of $Cl^-$/$SO_4^{2-}$, or $Li^+$/$Mg^{2+}$ of the SADs PA membranes.

The designed membranes displayed an ultrahigh $Cl^-$/$SO_4^{2-}$ selectivity, up to 2883 (for A-PA/BA-G4D) and 1652.6 (for A-PA/p-HC-G4D), which were higher than those of the control A-PA membrane and the commercial NF 270, XC-N, DK and DL membranes (Fig. 4a, b, Supplementary Tables 1 and 2). A trade-off chart between $Cl^-$/$SO_4^{2-}$ selectivity and water permeance was plotted to conveniently compare different membranes, including literature-reported data and this work (Fig. 4b). More than 130 sets of data were used for comparison, while most of the data associated with the literature reported had difficulty achieving a trade-off because of low selectivity or water permeance (Supplementary Tables 2 and 3). The SAD PA membranes exhibited a good performance trade-off, especially A-PA/p-HC-G4D and A-PA/BA-G4D. Specifically, due to the difference of pore structure and inner charge for A-PA/P-HC-G4D, A-PA/BA-G4D and A-PA/DA-G4D, the separation roles, including size sieving and the Donnan effect of these designed polyamide nanofilms on $Cl^-$/$SO_4^{2-}$ showed corresponding strength differences. For example, due to the larger $Cl^-$/$SO_4^{2-}$ efficient separation area for A-PA/p-HC-G4D (see below), its size sieving effect on the separation of $Cl^-$/$SO_4^{2-}$ was more significant than that of the Donnan effect on the separation of $Cl^-$/$SO_4^{2-}$ for A-PA/DA-G4D. In other words, when separating $Cl^-$/$SO_4^{2-}$, the synergetic effects of size sieving and the Donnan effect for A-PA/p-HC-G4D was better than that of A-PA/DA-G4D. Hence, A-PA/p-HC-G4D showed a higher $Cl^-$/$SO_4^{2-}$ separation factor than A-PA/DA-G4D. These results demonstrated that the prepared PA membranes have potential in the application process of zero liquid discharge for industry.

We further evaluated the SADs PA membranes when used in high LiCl and $MgCl_2$ concentrations, for example, in the process of extracting lithium from Salt Lake Brine. A series of $Li^+$ and $Mg^{2+}$ mass concentration ranges that basically matched with some of the brine lakes in the world were applied as feed solutions[29]. The separation performance results demonstrated that the $MgCl_2$ rejection rate was high >98.5%, even at a salt concentration of 15 g L$^{-1}$ (Fig. 4c). The LiCl rejection rate, however, gradually decreased with increasing salt concentration due to concentration polarization[12]. These data demonstrated that the SAD PA membranes have potential in the separation of $Li^+$ and $Mg^{2+}$ and stable performance at high salt concentrations.

On the other hand, the $Li^+$/$Mg^{2+}$ selectivity for the prepared PA membranes under different $Mg^{2+}$/$Li^+$ ratios (MLRs) and operational pressures was further assessed (Fig. 4d and Supplementary Table 4). Specifically, the MLR of 7.8 indicates that the mass concentrations of $Li^+$ and $Mg^{2+}$ are 81.87 ppm and 638.19 ppm, respectively. Similarly, an MLR of 15.6 indicates that the $Li^+$ and $Mg^{2+}$ mass concentrations are 81.87 ppm and 1276.38 ppm, respectively. An MLR of 31.2 exhibits mass concentrations of $Li^+$ and $Mg^{2+}$ of 81.87 ppm and 2552.75 ppm, respectively.

A significant $Li^+$/$Mg^{2+}$ selectivity enhancement was achieved while maintaining high $Mg^{2+}$ rejection (>98.5%) and, in many instances, even showing a negative $Li^+$ rejection rate relative to the control A-PA membrane (Supplementary Table 4). A-PA/BA-G4D showed excellent $Li^+$/$Mg^{2+}$ selectivity, up to 116.8 under an MLR of 31.2, and the $Mg^{2+}$ rejection was 98.53%, while that of $Li^+$ was −71.7%. For the control A-PA membrane under the same conditions, the $Mg^{2+}$ rejection rate was 90.65%, while that of $Li^+$ was −67.26%, and the obtained $Li^+$/$Mg^{2+}$ selectivity was 17.89. In regard to the negative rejection of $Li^+$, the ion rejection of mixed solution is generally impacted by the concentration of other ions and their difference among materialized parameters. The $Li^+$ concentration in the LiCl/$MgCl_2$ solution is far less than that in the $Cl^-$ (Supplementary Table 4). Taking 31.2 of MLR as an example, the total $Cl^-$ and $Mg^{2+}$ concentrations were 7889.3 ppm and 2552.75 ppm, respectively, whereas the $Li^+$ concentration was only 81.87 ppm. When

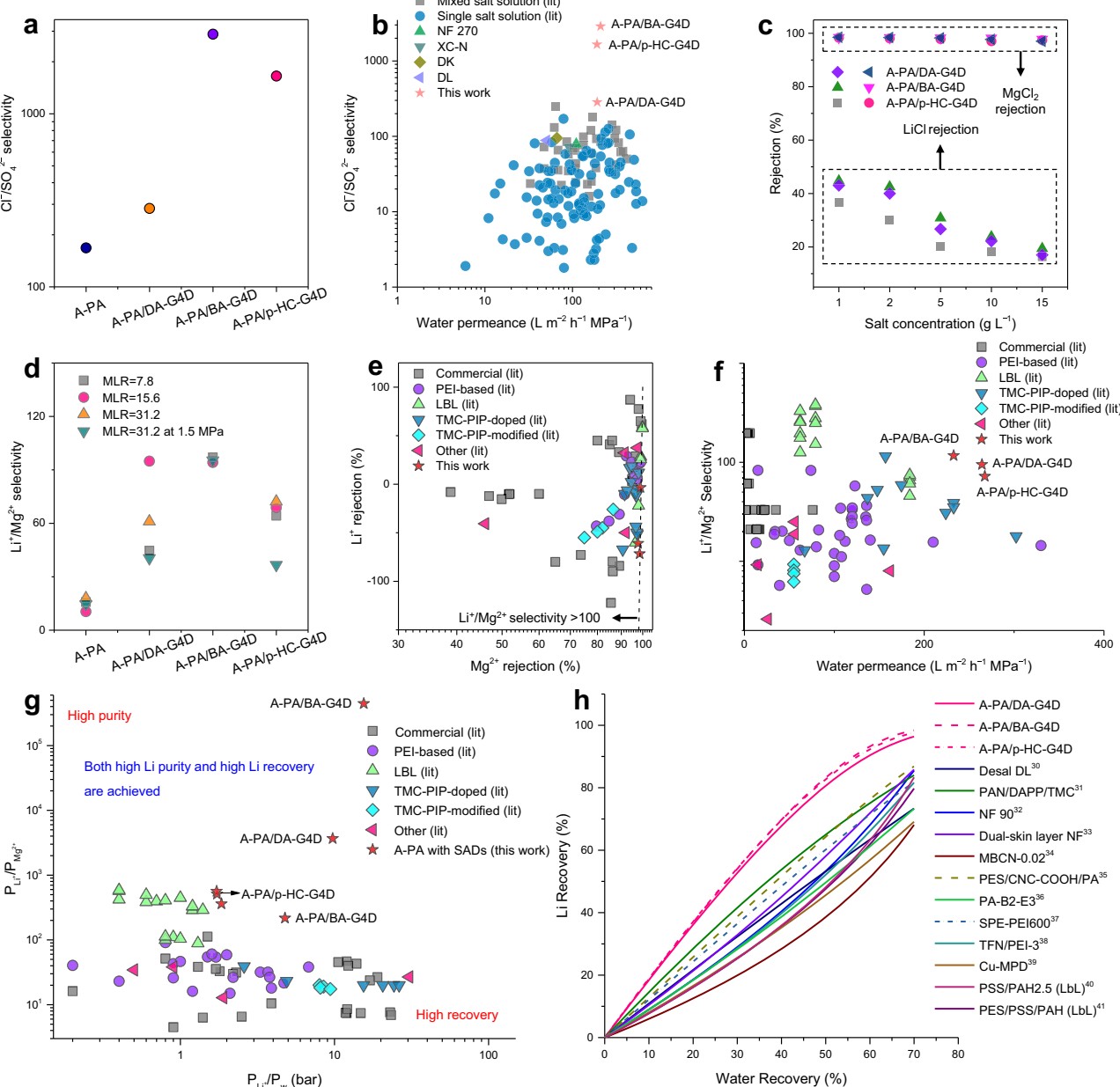

**Fig. 4 | Ion separation selectivity of the A-PA membrane and the SADs PA membranes (A-PA/DA-G4D, A-PA/BA-G4D, and A-PA/p-HC-G4D). a** $Cl^-/SO_4^{2-}$ selectivity tested under the feed solution of $2\,g\,L^{-1}$ NaCl and $2\,g\,L^{-1}$ Na$_2$SO$_4$. **b** Ion separation performance in terms of $Cl^-/SO_4^{2-}$ selectivity and water permeance of the resultant PA membranes (red) compared with literature data (blue and grey) and the commercial membranes (NF 270 and XC-N from DuPont, DK and DL from Suez). **c** LiCl and MgCl$_2$ rejection rates with different salt concentrations. **d** Li$^+$/Mg$^{2+}$ selectivity under different Mg$^{2+}$/Li$^+$ ratios (MLR) and operational pressures. **e** Li$^+$ rejection as a function of Mg$^{2+}$ rejection. **f** Ion separation performance in terms of Li$^+$/Mg$^{2+}$ selectivity and water permeance of the prepared SADs PA membranes

(red) compared with literature data. **g** Ratio for Li$^+$ permeability and Mg$^{2+}$ permeability ($P_{Li^+}/P_{Mg^{2+}}$) vs. ratio for Li$^+$ permeability and water permeability ($P_{Li^+}/P_w$). Increasing $P_{Li^+}/P_w$ enhances Li recovery, whereas increasing $P_{Li^+}/P_{Mg^{2+}}$ improves Li purity on the permeation side. The permeabilities are extracted using the solution-diffusion-electromigration (SDEM) model with an assumed mass transfer coefficient of $100\,L\,m^{-2}\,h^{-1}$ for LiCl and MgCl$_2$. **h** Module analysis: Calculated Li recovery vs. water recovery along a simulated module. Feed conditions: Li$^+$ concentration of 3.4 mM, Mg$^{2+}$ of 19.6 mM, and operating pressure of 4 bar. Membrane properties: choosing the best-reported data for each type of membrane[30,44–54]. The detailed data can be found in the Supplementary Information.

ions permeate through the PA membrane, there was always residual Mg$^{2+}$ that was not entirely intercepted in the permeation solution, making it possible that the Cl$^-$ concentration of the permeation side was larger than that of the initial 418.14 ppm from $0.5\,g\,L^{-1}$ LiCl. Therefore, due to the Donnan equilibrium effect, the Li$^+$ concentration on the permeation side was more likely to be even higher than that on the feed side. In addition, Li$^+$ has a more significant diffusion coefficient ($1.03 \times 10^{-9}\,m^2\,s^{-1}$, smaller hydrated radius (0.382 nm) and lower hydration energy ($474\,kJ\,mol^{-1}$) than Mg$^{2+}$ ($0.706 \times 10^{-9}\,m^2\,s^{-1}$,

0.428 nm, $1828\,kJ\,mol^{-1}$), making it easier and faster to pass through the nanofilm[29–32]. These Li$^+$/Mg$^{2+}$ selectivity outcomes suggest that, the SADs linked into the A-PA layer can enhance Mg$^{2+}$ rejection, while maintaining the permeation of Li$^+$.

Moreover, the SADs PA membranes possessed the same Li$^+$ and Mg$^{2+}$ rejection stability as the control A-PA membrane during the long-term filtration experiments, thus maintaining a high selectivity (Supplementary Fig. 44). The chart of Li$^+$ rejection as a function of Mg$^{2+}$ rejection in Fig. 4e explained that the designed SADs PA

membranes held a high $Mg^{2+}$ rejection, while revealing a considerably negative $Li^+$ rejection compared with other types of separation membranes. Expressly, most of the PEI-based membranes indicated both high $Mg^{2+}$ rejection and $Li^+$ rejection. In addition, scattered data points about the rejection rates of $Mg^{2+}$ and $Li^+$ for commercial PA membranes were observed (Fig. 4e). As confirmed in Supplementary Table 4 and Supplementary Fig. 45, since both excellent $Mg^{2+}$ rejection and low $Li^+$ rejection were obtained in mixed feed solution, and combined with the Equation 3 in the Supplementary Information, the high $Li^+/Mg^{2+}$ selectivity calculated was observed in this work. Further comparison to different membranes was conducted by plotting the chart on ion separation performance in terms of $Li^+/Mg^{2+}$ selectivity and water permeance trade-off. As demonstrated in Fig. 4f, most of the literature reported membranes, including commercial, PEI-based, layer-by-layer (LbL) and TMC–PIP-modified membranes, exhibited either low $Li^+/Mg^{2+}$ selectivity or low water permeance (Supplementary Tables 5 and 6), making it difficult to achieve a trade-off. However, both high selectivity and high water permeance were observed for the SADs PA membranes, showing an ability to break the current performance trade-off curve.

Based on the SDEM model[33–36], we fitted out the ion permeability using experimental data (Supplementary Software, Supplementary Tables 5 and 6) to further understand the advantages of the SADs PA membranes for $Li^+/Mg^{2+}$ separation. In real applications, high Li purity and high Li recovery are two main criteria related to the $Li^+/Mg^{2+}$ separation performance. The Li purity can be evaluated using the ratio of $Li^+$ permeability and $Mg^{2+}$ permeability ($P_{Li^+}/P_{Mg^{2+}}$), while the Li recovery can be referred to as the ratio of $Li^+$ permeability and water permeability ($P_{Li^+}/P_w$)[8]. As shown in Fig. 4g, the LbL membranes generally provided a high Li purity, while a comparably high Li recovery was observed for commercial membranes (DK, DL, NF 270, etc.). Our designed SADs PA membranes, especially the A-PA/DA-G4D and the A-PA/BA-G4D, exhibited a better trade-off of $P_{Li^+}/P_{Mg^{2+}}$ and $P_{Li^+}/P_w$ compared to the commercial products, the PEI-based, and the LbL type membranes. This result indicates that the strategy to generate an SAD PA membrane with a hollow nanostripe structure was effective, which improved the high Li purity while maintaining high Li recovery, as the membranes showed no loss in performance. Take the A-PA/BA-G4D membrane as an example. A $P_{Li^+}/P_{Mg^{2+}}$ up to 444273.94 was achieved, with an observed $P_{Li^+}/P_w$ of 15.5 bar. Details of the experimental conditions and the ionic composition of the feed and permeate samples, as well as the values of observed rejection of individual ions, are presented in Supplementary Table 5.

Furthermore, we applied a simulated module to calculate Li recovery vs. water recovery. Each membrane type with the best reported-data was chosen for comparison (Fig. 4h, Supplementary Fig. 46 and Table 7). The modelling result shows that, under the same pressure (4 bar and 10 bar) and mixed ion concentration (3.4 mM $Li^+$, 19.6 mM $Mg^{2+}$), a higher Li recovery was achieved with the A-PA/DA-G4D membrane using one element in series compared with other membranes, including commercial NF 90 and Desal DL, when producing the same water recovery. Moreover, as demonstrated in Supplementary Fig. 47, a simulated module on Li recovery vs. membrane area further suggested that if the same membrane area is applied in the vessel, a higher Li recovery may be achieved for the SADs polyamide membranes than other reported membranes. For example, for A-PA/p-HC-G4D, 40% Li recovery in the separation process may be achieved with a 1.69 $m^2$ membrane area, while for TFN/PEI-3, the membrane area used would be up to 3.48 $m^2$. In other words, the larger permeance of the SADs polyamide membranes would allow the use of a smaller membrane area to achieve the same Li recovery. In this case, a reduction in the power consumption from operation pressure would be achieved[16]. Moreover, with the scaling up production for these SADs polyamide membranes and the costs that synthesize these dendrimers would be decreased accordingly, a

potential advantage in terms of membrane replacement costs can also be expected.

## Molecular simulations and adsorption experiments

As shown in Fig. 5, a further inspection of the nanopore structure of PA networks containing SADs was conducted via computational models made by GROMACS v4.6.7, and all atoms were parameterized by the Gromos 54A7 force field[37–41]. Details about the molecular simulation are shown in the Methods section. Three models were produced as samples from different structural positions of each system and ensured the representativeness of the simulation results. It is known that the separation mechanism for the nanofiltration membranes mainly includes size sieving and the Donnan effect. Therefore, an efficient pore size range and uniform pore structure are significant for polyamide nanofilms in terms of ion separation. As demonstrated in Fig. 5a, PA networks incorporating SADs (PA/DA-G4D, PA/BA-G4D and PA/p-HC-G4D) have no pore structure >5 Å relative to the pristine PA polymer, exhibiting a narrower pore size range and a more uniform pore structure. The pore structure features are more conducive to conducting efficient ion separation. Moreover, as shown in Fig. 5c, PA-containing SAD polymer models offered improved simulated porosity (37.38% to 38.29%) relative to the pure PA model (36.35%), which might be due to the dendrimers featuring inherent micropores (Supplementary Fig. 48). The PA/SADs also exhibited an increased simulated density (Fig. 5d), which was consistent with the volumetric density calculated in Fig. 3a. This was because of the incorporation of dendrimers with the full-aromatic chemical structure, whereas that of PA was semi-aromatic structure, which increased the density of the final polyamide polymer[42].

More importantly, the simulated pore size distributions averaged in Fig. 5b further demonstrate that a uniform distribution of ultra-small pores was observed for the PA-containing SAD polymer compared to pure PA. Specifically, the PA has an apparently broader distribution range of ultrasmall pores between 1.7–9.8 Å, whereas that of PA/DA-G4D is 1.8–4.6 Å, 1.5–4.4 Å for PA/BA-G4D, and 1.8–4.1 Å for PA/p-HC-G4D. It is known that a uniform distribution of ultrasmall pores could achieve the effective sieving of ions[12]. Here, we define the pore range between 3.82 Å and 4.28 Å as the ideal $Li^+/Mg^{2+}$ separation area, and the pore range between 3.32 Å and 3.79 Å as the ideal $Cl^-/SO_4^{2-}$ separation area (Supplementary Table 8). Closer observation found that, compared with the pristine PA polymer, the PA incorporating SADs displays an enhanced effective separation area for $Li^+/Mg^{2+}$. The integral area calculated for PA is 0.10, while that of 0.14 for PA/DA-G4D, 0.16 for PA/BA-G4D and 0.17 for PA/p-HC-G4D. The effective separation area increase facilitates $Li^+$ transport through the nanofilm and suppresses $Mg^{2+}$ permeation, showing enhanced $Li^+/Mg^{2+}$ permselectivity. This result is also consistent with comparable experimental $Li^+/Mg^{2+}$ separation data. The separation area for $Cl^-/SO_4^{2-}$ was enhanced with the incorporation of SADs as well, which matched the $Cl^-/SO_4^{2-}$ separation outcomes.

On the other hand, as shown in Fig. 5e, since a uniform and narrow distribution of pore structure was observed for the PA-containing SADs polymer compared to the pristine PA polymer, it showed a lower NaCl adsorption content. This difference in the NaCl adsorption of the pristine A-PA nanofilm and the SADs PA nanofilms further confirmed the link between the chemical structure and ion separation performance. The simulation and NaCl adsorption experimental results gave a relatively more uniform pore structure, higher porosity and more effective ion separation area for A-PA/DA-G4D, A-PA/BA-G4D and A-PA/p-HC-G4D over the control A-PA polymer networks.

## Discussion

Positively charged SADs with different morphologies were formed in PIP solution and used as an aqueous-phase amine reaction system, and we successfully fabricated SAD asymmetric polyamide nanofilms with

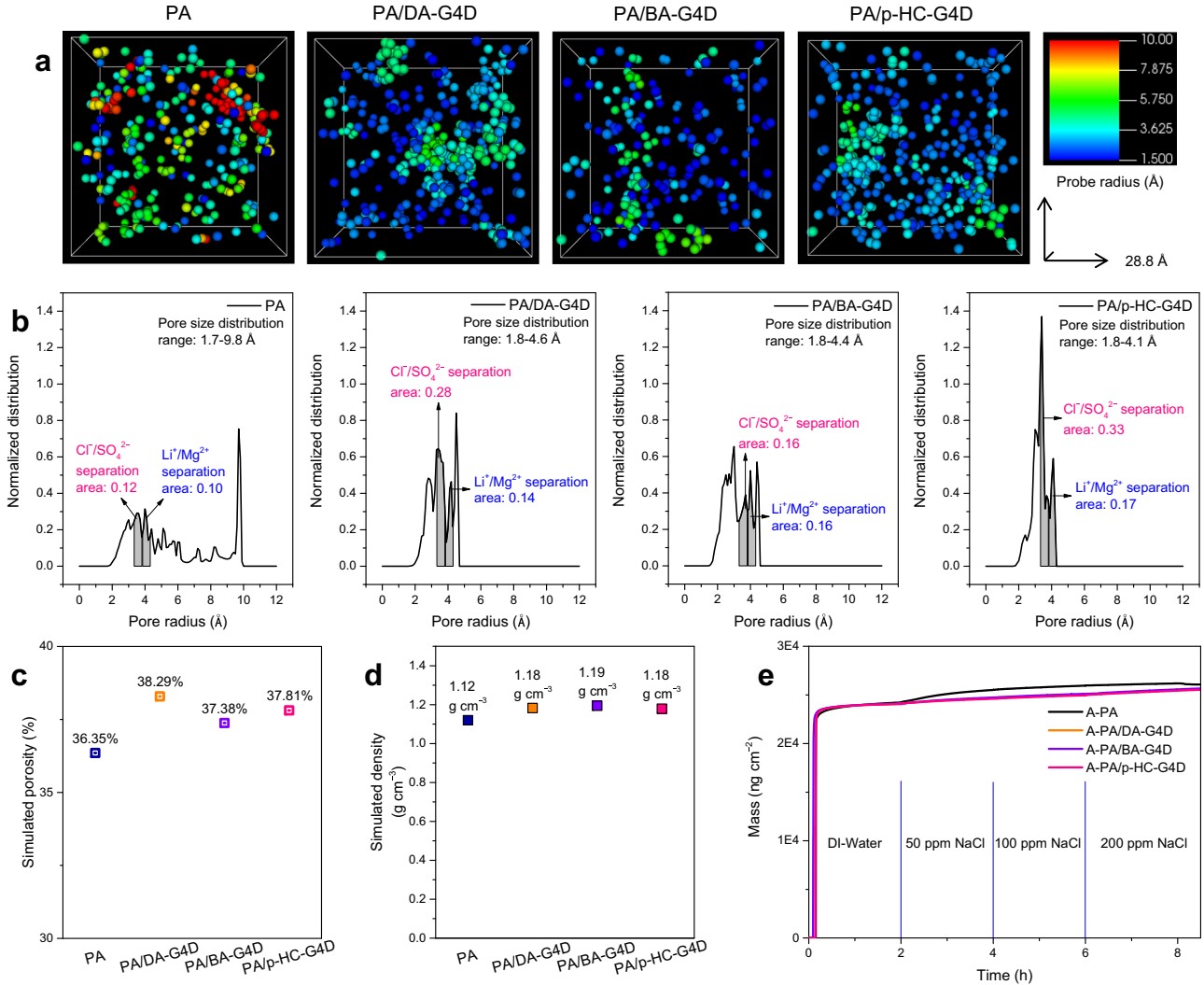

**Fig. 5 | Molecular simulations and adsorption experiments. a** Representative pore distribution images with respect to a probe with a 1 Å radius for the PA polymer incorporating SADs compared with the pristine PA polymer. The right row shows the probe radius with different colours. **b** Simulated pore size distributions averaged and relevant pore size range for the four PA systems. Shades of grey sections means the calculated theoretical ion separation area. **c**, **d** Simulated porosity and density for the resultant nanofilms. **e** Adsorption behaviours of NaCl, from 50 ppm to 200 ppm, on the formed nanofilms monitored by QCM as a function of time.

hollow nanostripe structures. The experimental and simulation results obtained here demonstrate that the SAD polyamide nanofilms provide an enhanced effective separation pore range and uniform pore structure. These optimized membranes allow the larger permselectivity of Li$^+$ over Mg$^{2+}$ and faster permeation of Li$^+$ over water compared with the market and other types of literature-reported membranes, achieving high Li purity and recovery. In addition, the optimized SADs PA membranes show a better trade-off between Cl$^-$/SO$_4^{2-}$ selectivity and water permeance. It also demonstrates excellent structural compatibility for PA segments and SADs, achieved by forming SADs in aqueous-phase PIP solutions. This work promotes the application of a self-assembled aqueous-phase solution in terms of interfacial polymerization to develop next-generation efficient ion, molecule, and desalination nanofilms.

## Methods

### Synthesis of the DA-G4D dendrimer
G4 dendrimer (G4D) was synthetized according to the Supplementary Methods[24,43]. Diethanolic anhydride (DA) (0.324 g, 2.792 mmol) was added to the dissolved G4D solution (0.3 g, 0.0727 mmol, 10 mL NMP) for reaction in an ice water bath for 30 min. After being placed at room temperature for 2 h, deionized water (8.375 µL, 0.465 mmol) was added, and the system temperature was raised to 50 °C for a further reaction of 2 h. The obtained solution was reprecipitated in 100 mL of water and centrifuged to obtain the crude product. Finally, freeze-drying was conducted to obtain a pure DA-G4D product (89.45%, 0.510 g).

The chemical structure of the DA-G4D dendrimer and its spectroscopic data are shown in the Supplementary Information.

### Synthesis of the BA-G4D dendrimer
1, 2, 4-Benzeneticarboxylic anhydride (BA) (0.447 g, 2.326 mmol) was added to the dissolved G4D solution (0.25 g, 0.0606 mmol, 10 mL NMP) for reaction in an ice water bath for 30 min. After incubation at room temperature for 2 h, deionized water (6.962 µL, 0.387 mmol) was added, and the system temperature was raised to 50 °C for a further reaction of 2 h. The final solution was reprecipitated in 100 mL of water and centrifuged to obtain the crude product. Finally, freeze-drying was conducted to obtain a pure DA-G4D product (80.76%, 0.503 g).

The chemical structure of the BA-G4D dendrimer and its spectroscopic data are shown in the Supplementary Information.

## Synthesis of the p-HC-G4D dendrimer

p-Hydroxybenzoic acid (2.5 g, 0.018 mol) was dissolved in 15 mL of benzene and 5 mL of N, N-dimethylformamide (DMF). Thionyl chloride (4.25 g, 0.036 mol) was added dropwise to the above solution within 30 min, under reflux and stirring, at 30 °C. After that, a further reaction proceeded at 45 °C for 4 h. The obtained solution was cooled to room temperature and transferred to a separatory funnel to obtain the lower layer solution. Vacuum distillation was conducted to remove sulfoxide chloride, benzene, and DMF and obtain a viscous solid (crude product). The crude product formed a supersaturated solution of dichloromethane and was recrystallized in n-hexane to obtain p-hydroxybenzoyl chloride (p-HC) (91.50%, 2.593 g).

p-HC (0.437 g, 2.792 mmol) was added to the dissolved G4D solution (0.3 g, 0.0727 mmol, 10 mL NMP) for reaction in an ice water bath for 30 min. After being placed at room temperature for 2 h, deionized water (8.375 μL, 0.465 mmol) was added, and the system temperature was raised to 50 °C for a further reaction of 2 h. The final solution was reprecipitated in 100 mL of water and centrifuged to obtain the crude product. Finally, freeze-drying was conducted to obtain a pure p-HC-G4D product (79.21%, 0.459 g).

The chemical structures and spectroscopic data of p-hydroxybenzoyl chloride (p-HC) and the p-HC-G4D dendrimer are shown in the Supplementary Information.

## Fabrication of SADs aqueous-phase PIP solution

The prepared functional dendrimers were dissolved in NMP, and then added dropwise into the PIP solution. Self-assembled dendrimers were obtained after standing and then formed the SAD aqueous-phase PIP solution, which was used to react with TMC/cyclohexane solution. The formation of SADs solution is related to the concentrations of dendrimers and PIP solutions.

## Fabrication of an SAD polyamide nanofilm with an asymmetric nanostripe structure

A dendrimer porous layer on the PSF ultrafiltration (UF) membrane was formed according to our previous report[23]. Under the optimized SAD aqueous-phase PIP solution, SAD polyamide nanofilms with asymmetric nanostripe structures were formed. Specifically, the concentrations of the optimized DA-G4D, BA-G4D and p-HC-G4D SADs PIP solutions were 0.04 wt.% DA-G4D and 0.8 wt.% PIP, 0.04 wt.% BA-G4D and 0.8 wt.% PIP, and 0.08 wt.% p-HC-G4D and 0.8 wt.% PIP, respectively. The UF membrane containing the dendrimer porous layer was first immersed into the above SAD water-phase PIP solution for 2 min. Subsequently, the extra SAD solution on the support surface was blown off with an air knife. Then, the above support was contacted with TMC/cyclohexane solution (0.1 w/v%) for 30 s to form the SAD polyamide membrane. The resultant polyamide membranes were washed with n-hexane for 15 s, finally dried at 60 °C for 2 min and stored in DI water until use. For the control samples, 0.8 wt.% PIP was applied, and conducted the same fabrication process was conducted to form the control polyamide membrane.

## Molecular simulations

In this work, the B3LYP function in the quantum mechanics software Gaussian 09, Revision B.01, was used to calculate the charge of each atom of TMC and PIP. In addition, during the DFT calculations, 6-31 + g (d, p) basis functions were applied. After removing one hydrogen atom and one chlorine atom from the amino and acyl chloride groups of the two monomers, the net charges were −2 and +3, respectively. Then, all atoms were parameterized by Gromos 54A7 force field and Automated Topology Builder (ATB), such as bond length, bond angle and dihedral angle.

First, the crosslinking process of monomers was simulated through MD. By adjusting the PIP/TMC monomer ratio to 3:2, the net charge of the entire system was ensured to be 0. Twelve PIP molecules and eight TMC molecules were randomly inserted into a simulated box with a side length of 2.5 nm and then dynamically crosslinked for 4 ns in vacuum under the NVT (300 K) ensemble (constant molecular number, constant volume, constant temperature). After the completion of the simulated crosslinking, the amide bonds in the formed molecular structure were connected, where the distance between C and N should be >0.35 nm. A small PA polymer segment with a cross-linked structure was then obtained with a simulated box size of 2.345 × 2.345 × 2.345 nm³. The crosslinking degree of this small PA segment was 83.3%, which was consistent with the experimental measurement value. Afterwards, this small PA segment was amplified twice by replicating itself in the X, Y, and Z directions. Finally, a very thin film model was replicated twice along the Z direction until a relatively uniform PA film with a thickness of 8.33 nm was formed.

For each step of amplification, the steepest descent algorithm was used for system energy minimization, and three-dimensional periodic boundary conditions (PBC) were applied. During the simulation process, to make the structure of the polymer film more uniform, a heating annealing algorithm was applied, and the simulation time was as long as 120 ns. The annealing temperature varied from 300 K to 500 K. After a series of repeated prolonged heating and annealing processes, the orientation of the PA chains became random, which was consistent with the true amorphous structure of PA. The PA polymer model processed by the annealing algorithm exhibited a density of approximately 1.1197 g cm⁻³, which was in good agreement with the experimental value of 1.119 g cm⁻³. For the other polymer models containing DA-G4D, BA-G4D and p-HC-G4D dendrimers, the construction process was similar to the above steps.

For the process of swelling the PA membrane, a fixed size box was first established, then the membrane was adjusted to the centre of the box, and finally the entire simulated box was filled with water molecules. The initial configuration of the system was as follows: Each system should run an equilibrium simulation of 20 ns under an NPT ensemble (constant molecular number, constant pressure, constant temperature). During the simulation process, the temperature was maintained at 298.15 K, and the pressure remained constant at 1 bar atmospheric pressure.

All MD simulations were conducted in GROMACS v4.6.7 software with a time step of 2 fs. Before conducting MD calculations, the steepest descent method was applied to minimize the initial energy of each system, with a force tolerance of 1 kJ/(mol⁻¹ nm⁻¹) and a maximum step size of 1 fs. The speed allocated by the preprocessor was based on the Maxwell Boltzmann distribution at 300 K. During the simulation process, the Gromos 54A7 force field was used to describe the molecules in the system. H₂O was described by the SPC (simple water models) model. Periodic boundary conditions (PBC) were applied in all three directions. The Leapfrog algorithm was used to integrate the motion equations of Newton. In addition, during the NPT simulation process, the isotropic Berendsen method was used to maintain the pressure at 1 bar. The temperature was maintained at the specified temperature by a V-rescale type thermostat. The particle mesh Ewald method with fourth-order interpolation was applied as the method for evaluating electrostatic interactions, of which the grid spacing was 1.0 Å and the truncation radius was 1.0 Å. Using this step, the Lennard Jones (LJ) interaction was successfully calculated. The gro file of the results was displayed using VMD (Visual Molecular Dynamics) visualization software.

Experimental details of the characterization methods, membrane performance test, density measurement and adsorption behaviours are illustrated in the Supplementary Information.

## Data availability

The figure data generated in this study are provided in the Supplementary Information/Source Data file. The data that support the

findings of this study are also available from the corresponding author upon request. Source data are provided with this paper.

## Code availability

All codes written for and used in this study to generate Fig. 4g, h are available publicly from Supplementary Software. Matlab Codes for the calculation of $P_{Li^+}$ and $P_{Mg^{2+}}$ can be accessed in Supplementary Information.

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

## Acknowledgements

The National Natural Science Foundation of China (Grant Nos. 22008056, 22111530288, 41907121, 22308090 and U2006230), Henan Province University Science and Technology Innovation Talent Support Plan (24HASTIT005), Henan Provincial Science and Technology R&D Program Joint Fund Project (222301420044), and Application Research Plan for Key Scientific Research Projects in Higher Education Institutions of Henan Province (23A150010) provided financial support for this work. We are also gratefully acknowledged to the Youth Science Foundation of Henan Normal University (Grant No. 2021QK11) for providing funding support. We would like to thank Y. W. Qin and Y. X. Hu from the State Key Laboratory of Separation Membranes and Membrane Processes for adsorption experiments, P. Y. Xin and L. L. Mao from State Key Laboratory of Antiviral Drugs for High Resolution Mass Spectrometer.

## Author contributions

B.Y. and Q.J.N. designed the research and proposed the idea. Y.Z., B.Y. and S.Z. synthesized the dendrimers, fabricated the membranes, the performed SEM, HRTEM, TEM, DLS, and zeta characterization and analysis. Y.Z. and K.Z. performed the separation tests. D.Y. conducted the module analysis in MATLAB. P.Q. performed the molecular simulations. P.H., X.Z., M.Y., J.C., J.J. and X.L. conducted the experimental analysis. B.Y. wrote the manuscript with input from all authors. All authors discussed the results and commented on the manuscript.

## Competing interests

The authors declare no competing interests.
