## [Peer review file · Nature Communications]

REVIEWER COMMENTS

Reviewer #1 (Remarks to the Author):

NCOMMS-23-25750

Comments to the Author

This study synthesized functionalized dendrimers (DA-G4D, BA-G4D, p-HC-G4D) with different functional groups that preferentially self-assembled in PIP solution to facilitate the formation of polyamide nanofilms with well-defined effective pore range. Compared with the pristine PA membrane, the self-assembled dendrimers (SADs) polyamide nanofilms achieved a high Cl⁻/SO₄²⁻ selectivity more than 17 times as compared with the pristine polyamide. The separation of Li⁺/Mg²⁺, achieving higher Li purity and recovery.

I do recommend publishing in Nature communication after major revision.

1: The separation factor in Supplementary Table 1 and Supplementary Table 4 is not consistent with the Supplementary Equation 3. In Equation 3 is α , but in the table is S. Please modified.

2: Why the model analysis pressure in Figure 4h is 4 bar, but 1 MPa is selected in experiment part. After 70 % water recover, is the osmotic pressure of the salt solution higher than 4 bar ?

3: Page 15, Line 335-338. Why is Li⁺ ion negatively intercepted? There is no explanation here.

4: Page 9, Line 3197-199: 'The upper surfaces images from SEM showed the existence of SADs (DA-G4D, BA-G4D and p-HC-G4D) nanoparticles covalently linked within the surrounding of the nanostripe PA layer (Figs. 2b-2d, Supplementary Figs. 18-20).' It is difficult to judge covalently linked from SEM. Please explain.

5: ①Page 7, Line 168-170: 'The BA-G4D and p-HC-G4D SADs exhibit a negative surface charge under a 0.01 wt.% PIP and 0.01 wt.% dendrimer (Supplementary Figs. 16a-16c).' However, the concentrations in Supplementary Figs. 16a-16c were not 0.01 wt.% PIP and 0.01 wt.% dendrimer.

②Page 7, Line 170-174: 'With the increased concentration (0.8 wt.% PIP and 0.04 wt.% dendrimer), the DA-G4D, BA-G4D and p-HC-G4D SADs possess a positive surface charge, which indicates that the peripheries of SADs are covered with PIP molecules and can be involved in the IP to form amide bond, and fine-tune the nanofilm inner structure (Supplementary Figs. 16d-16f).' But this sentence is not a description of Supplementary Figs.16d-16f. Please modify the description of Supplementary Figs. 16.

6: What is the concentration of PIP and synthesized functionalized dendrimers (DA-G4D, BA-G4D, p-HC-G4D) in Figure 3 and Figure 2?

7. In Figure 1: What state was the SADs in PIP water phase? In single particle or aggregated state.

8. The SADs (DA-G4D, BA-G4D and p-HC-G4D) have dendrimers structure. The structure shows large voids. What is the diameter of the voids? These voids allow passage of water. Do they also allow passage of ions?

9. How to calculate the volumetric charge densities according to zeta potential values in Figure 3a?

10. Figure 5 c, d, e does not correspond to the legend description.

Reviewer #2 (Remarks to the Author): Recommendation: Reconsider after major changes

Yuan et al. reported the synthesis of polyamide membranes modified with dendrimers and their use in the separation of ions of similar sizes. In their investigation, the authors hypothesized that dendrimers could be used to modify the pore volume of polyamide membranes to make them more selective for ion separation. The authors synthesized an amine-terminal dendrimer and functionalized it to make it water-soluble. The authors fabricated membranes using dendrimers and polyamides on a polysulfone support, which were characterized and used in ion rejection experiments. The authors performed a series of comparisons with a control membrane and commercial membranes to demonstrate the superior performance of dendrimer/polyamide membranes. Simulations were performed to support the differential ion rejection findings.

The following general points must be addressed for the manuscript to be considered for publication.

1. The manuscript requires grammar improvement to deliver a clearer message to the audience. Some areas of improvement are addressed in the specific comments.
2. The manuscript requires a better organization of the figures, especially in the supporting information. The authors need to consider a reorganization that makes the text reading more congruent and organized with the figures presented.
3. This manuscript has many claims that are not supported by the results. Authors must remove some of the claims or perform experiments that support them.
4. Although the authors tested three different membranes, there is no clear rationale for why one is better than the others. The authors need to include an analysis that correlates the properties of the different dendrimers with the rejection study results.

The following are specific comments:

1. Line 45-47: Consider rephrasing by combining the two phrases to form a sentence.
2. Line 50-51: Hard to follow and consider rephrasing.
3. Line 133-135: Do the authors have DLS data to support the HRTEM findings?
4. Line 139: What image?
5. Line 140-142: Consider rephrasing.
6. Line 156-159: Any suggestion why DA-G4D behaves differently?
7. Line 159-161: There is a mismatch between the figures and the concentrations.
8. Line 161-163: What is the final concentration of dendrimers and PIP? Why are all dendrimers single nanoparticles?
9. Line 174-177: Which ones are the best conditions? What are the criteria for their selection?
10. Line 181-183: I do not follow the discussion; what does the authors mean by the inconsonant diffusion-reaction rate?
11. Line 183-186: It is not clear how the polyamide film is formed. What is the amount of PIP required to fabricate the polyamide film? For the control (A-PA) how much PIP was used? Do the authors use another diamine to form the polyamide films?
12. Line 188-190: The authors claim a covalent bond between the SADs and the PA film. However, no experiment has supported this claim. Is the bond between the dendrimers and PIP covalent? How do they know that PIP is covalently attached to PA films?

13. Line 1998: SEM cannot be used to claim a covalent linkage between the dendrimers and the PA film.
14. Line 213-218: It is not clear to me what is the relationship between layer thickness and peripheral PIP molecules. If dendrimers with a lower amount of PIP were to be used, the thickness of the PA film would be higher? What are the errors in thickness values?
15. Line 218-221: Consider rephrasing.
16. Line 221: Why is it clear that nanovoids are present?
17. Line 223-227: Authors claim that dendrimers have a negative charge. This claim seems to be true when 0.01% SADs concentration was used. However, when 0.04 and 0.08 % SADs were used, the charge was positive. In the methods section (line 508), the authors stated that 0.04 % and 0.08 % SADs concentrations were used. Why the discrepancy?
18. Line 224: How do the authors know that there is a covalent bond?
19. Line 247-248: What is the importance of membranes with similar wettability?
20. Line 267-271; Consider discussing the water flux for LiCl as well.
21. Line 278-279: Authors mention optimized conditions, but do not comment what are those conditions and how they obtained them.
22. Line 308-309: Authors should provide a context for why the studied ions are important.
23. Line 319-321: What is the rationale that makes A-PA/P-HC-G4D and A-PA/BA-G4D better than A-PA/DA-G4D?
24. Line 322-325: The authors used the same concentration for both Li and Mg for the high concentration studies. A more relevant investigation would be to use the average concentrations of these ions in salt-lake brines. How common are the selected concentrations used in this section?
25. Line 329-330: What are the initial concentrations of Li and Mg in this section? This information is important for understanding the relevance of this study.
26. Line 324: The y-axis of supplementary Fig. 33 is not selectivity, and it is not for Li/Mg; the authors need to update the figure.
27. Line 348: I recommend changing the x-axis of Fig. 4e to a log scale to obtain a better visualization of the data.
28. Line 348-350: Consider rephrasing.
29. Line 352-353: What is the rationale for adding the upper bond line? If this is an arbitrary decision, it is recommended to remove it. If is not an arbitrary decision, I recommend providing context.
30. Line 371: What is the rationale for adding an upper bond line? If this is an arbitrary decision, it is recommended to remove it. If is not an arbitrary decision, I recommend providing context.
31. Line 388-392: Authors need to include a cost analysis to claim that their membrane is cheaper than commercial membranes.
32. Line 404-405: Phrase does not seem connected to the rest of the paragraph. Consider rephrasing.
33. Line 408-410: I do not follow the discussion in this sentence. Are the authors discussing the importance of the size, uniformity, or both? Therefore, rephrasing is recommended.
34. Line 410-417: It is difficult for me to understand how a material can be simultaneously more porous and denser. Seems to me that these two features are inversely proportional.
35. Line 435: What is the difference in the NaCl adsorption?

Reviewer #3 (Remarks to the Author):

This paper deals with precision separation of mono/di-valent ions. Specifically Li⁺/Mg²⁺ separation (and associated Cl⁻/SO₄²⁻) separation. The topic is important as resource recovery (as exemplified in the concept of 'brine mining' is attracting more and more attention - and seen as an important aspect of achieving circularity in resource management.

The paper describes synthesis, characterization and functional tests of membranes based on dendrimers with carboxyl and phenolic hydroxyl groups embedded in a polyamide structure.

Overall the paper builds on a sound methodology. The experimental results are generally well presented and impressive Li⁺/Mg²⁺ selectivities (from the various dendrimer-based embodiments) are achieved. This is indeed a promising result in the development of membranes for Li⁺-recovery.

There are however a few issues which need to be addressed. First, the use of dendrimer-based membranes for ion separation is not new - and citations to recent papers (e.g. Z-L, Qiu et al. ACS Nano 15, 7522-7535, 2021) are missing. Second, for the modeling work: why was the B3LYP functional chosen? There are other DFT functionals each with advantages and disadvantages (for a nice review see N. Mardirossian & M. Head-Gordon. MOLECULAR PHYSICS, 2017, 115(19), 2315-2372). Third, what role (if any) does electronic polarization of the constituent molecular structure play in the separation mechanism and how is this addressed in the MD simulations? Finally, how does the MD PBC setup (which is indeed a good starting point) reflect the true corrugated membrane surface - and what implications does this simplification have for the results presented?

Reviewer: 1

Reviewer #1 Comments to the Author

This study synthesized functionalized dendrimers (DA-G4D, BA-G4D, p-HC-G4D) with different functional groups that preferentially self-assembled in PIP solution to facilitate the formation of polyamide nanofilms with well-defined effective pore range. Compared with the pristine PA membrane, the self-assembled dendrimers (SADs) polyamide nanofilms achieved a high Cl⁻/SO₄²⁻ selectivity more than 17 times as compared with the pristine polyamide. The separation of Li⁺/Mg²⁺, achieving higher Li purity and recovery.

I do recommend publishing in Nature communication after major revision.

1: The separation factor in Supplementary Table 1 and Supplementary Table 4 is not consistent with the Supplementary Equation 3. In Equation 3 is α , but in the table is S. Please modified.

2: Why the model analysis pressure in Figure 4h is 4 bar, but 1 MPa is selected in experiment part. After 70 % water recover, is the osmotic pressure of the salt solution higher than 4 bar?

3: Page 15, Line 335-338. Why is Li⁺ ion negatively intercepted? There is no explanation here.

4: Page 9, Line 3197-199: 'The upper surfaces images from SEM showed the existence of SADs (DA-G4D, BA-G4D and p-HC-G4D) nanoparticles covalently linked within the surrounding of the nanostripe PA layer (Figs. 2b-2d, Supplementary Figs. 18-20).' It is difficult to judge covalently linked from SEM. Please explain.

5: ①Page 7, Line 168-170: 'The BA-G4D and p-HC-G4D SADs exhibit a negative surface charge under a 0.01 wt.% PIP and 0.01 wt.% dendrimer (Supplementary Figs. 16a-16c).'

However, the concentrations in Supplementary Figs. 16a-16c were not 0.01 wt.% PIP and 0.01 wt.% dendrimer.

②Page 7, Line 170-174: 'With the increased concentration (0.8 wt.% PIP and 0.04 wt.% dendrimer), the DA-G4D, BA-G4D and p-HC-G4D SADs possess a positive surface charge, which indicates that the peripheries of SADs are covered with PIP molecules and can be involved in the IP to form amide bond, and fine-tune the nanofilm inner structure (Supplementary Figs. 16d-16f).' But this sentence is not a description of Supplementary Figs.16d-16f.

Please modify the description of Supplementary Figs. 16.

6: What is the concentration of PIP and synthesized functionalized dendrimers (DA-G4D, BA-G4D, p-HC-G4D) in Figure 3 and Figure 2?

7. In Figure 1: What state was the SADs in PIP water phase? In single particle or aggregated state.

8. The SADs (DA-G4D, BA-G4D and p-HC-G4D) have dendrimers structure. The structure shows large voids. What is the diameter of the voids? These voids allow passage of water. Do they also allow passage of ions?

9. How to calculate the volumetric charge densities according to zeta potential values in Figure 3a?

10. Figure 5 c, d, e does not correspond to the legend description.

Reviewer: 2

Reviewer #2 (Remarks to the Author):

Recommendation: Reconsider after major changes

Yuan et al. reported the synthesis of polyamide membranes modified with dendrimers and their use in the separation of ions of similar sizes. In their investigation, the authors hypothesized that dendrimers could be used to modify the pore volume of polyamide membranes to make them more selective for ion separation. The authors synthesized an amine-terminal dendrimer and functionalized it to make it water-soluble. The authors fabricated membranes using dendrimers and polyamides on a polysulfone support, which were characterized and used in ion rejection experiments. The authors performed a series of comparisons with a control membrane and commercial membranes to demonstrate the superior performance of dendrimer/polyamide membranes. Simulations were performed to support the differential ion rejection findings. The following general points must be addressed for the manuscript to be considered for publication.

1. The manuscript requires grammar improvement to deliver a clearer message to the audience. Some areas of improvement are addressed in the specific comments.

2. The manuscript requires a better organization of the figures, especially in the supporting information. The authors need to consider a reorganization that makes the text reading more congruent and organized with the figures presented.

3. This manuscript has many claims that are not supported by the results. Authors must remove some of the claims or perform experiments that support them.

4. Although the authors tested three different membranes, there is no clear

rationale for why one is better than the others. The authors need to include an analysis that correlates the properties of the different dendrimers with the rejection study results.

The following are specific comments:

1. Line 45-47: Consider rephrasing by combining the two phrases to form a sentence.

2. Line 50-51: Hard to follow and consider rephrasing.

3. Line 133-135: Do the authors have DLS data to support the HRTEM findings?

4. Line 139: What image?

5. Line 140-142: Consider rephrasing.

6. Line 156-159: Any suggestion why DA-G4D behaves differently?

7. Line 159-161: There is a mismatch between the figures and the concentrations.

8. Line 161-163: What is the final concentration of dendrimers and PIP? Why are all dendrimers single nanoparticles?

9. Line 174-177: Which ones are the best conditions? What are the criteria for their selection?

10. Line 181-183: I do not follow the discussion; what does the authors mean by the inconsonant diffusion-reaction rate?

11. Line 183-186: It is not clear how the polyamide film is formed. What is the amount of PIP required to fabricate the polyamide film? For the control (A-PA) how much PIP was used? Do the authors use another diamine to form the polyamide films?

12. Line 188-190: The authors claim a covalent bond between the SADs and the PA film. However, no experiment has supported this claim. Is the bond between the dendrimers and PIP covalent? How do they know that PIP is covalently attached to PA films?

13. Line 1998: SEM cannot be used to claim a covalent linkage between the dendrimers and the PA film.

14. Line 213-218: It is not clear to me what is the relationship between layer thickness and peripheral PIP molecules. If dendrimers with a lower amount of PIP were to be used, the thickness of the PA film would be higher? What are the errors in thickness values?

15. Line 218-221: Consider rephrasing.

16. Line 221: Why is it clear that nanovoids are present?

17. Line 223-227: Authors claim that dendrimers have a negative charge.

This claim seems to be true when 0.01% SADs concentration was used. However, when 0.04 and 0.08 % SADs were used, the charge was positive. In the methods section (line 508), the authors stated that 0.04 % and 0.08 % SADs concentrations were used. Why the discrepancy?

18. Line 224: How do the authors know that there is a covalent bond?

19. Line 247-248: What is the importance of membranes with similar wettability?

20. Line 267-271; Consider discussing the water flux for LiCl as well.

21. Line 278-279: Authors mention optimized conditions, but do not comment what are those conditions and how they obtained them.

22. Line 308-309: Authors should provide a context for why the studied ions are important.

23. Line 319-321: What is the rationale that makes A-PA/P-HC-G4D and A-PA/BA-G4D better than A-PA/DA-G4D?

24. Line 322-325: The authors used the same concentration for both Li and Mg for the high concentration studies. A more relevant investigation would be to use the average concentrations of these ions in salt-lake brines. How common are the selected concentrations used in this section?

25. Line 329-330: What are the initial concentrations of Li and Mg in this section? This information is important for understanding the relevance of this study.

26. Line 324: The y-axis of supplementary Fig. 33 is not selectivity, and it is not for Li/Mg; the authors need to update the figure.

27. Line 348: I recommend changing the x-axis of Fig. 4e to a log scale to obtain a better visualization of the data.

28. Line 348-350: Consider rephrasing.

29. Line 352-353: What is the rationale for adding the upper bond line? If this is an arbitrary decision, it is recommended to remove it. If is not an arbitrary decision, I recommend providing context.

30. Line 371: What is the rationale for adding an upper bond line? If this is an arbitrary decision, it is recommended to remove it. If is not an arbitrary decision, I recommend providing context.

31. Line 388-392: Authors need to include a cost analysis to claim that their membrane is cheaper than commercial membranes.

32. Line 404-405: Phrase does not seem connected to the rest of the paragraph. Consider rephrasing.

33. Line 408-410: I do not follow the discussion in this sentence. Are the

authors discussing the importance of the size, uniformity, or both? Therefore, rephrasing is recommended.

34. Line 410-417: It is difficult for me to understand how a material can be simultaneously more porous and denser. Seems to me that these two features are inversely proportional.

35. Line 435: What is the difference in the NaCl adsorption?

Reviewer: 3

Reviewer #3 (Remarks to the Author):

This paper deals with precision separation of mono/di-valent ions. Specifically Li⁺/Mg²⁺ separation (and associated Cl⁻/SO₄²⁻) separation. The topic is important as resource recovery (as exemplified in the concept of 'brine mining' is attracting more and more attention - and seen as an important aspect of achieving circularity in resource management.

The paper describes synthesis, characterization and functional tests of membranes based on dendrimers with carboxyl and phenolic hydroxyl groups embedded in a polyamide structure.

Overall the paper builds on a sound methodology. The experimental results are generally well presented and impressive Li⁺/Mg²⁺ selectivities (from the various dendrimer-based embodiments) are achieved. This is indeed a promising result in the development of membranes for Li⁺-recovery.

There are however a few issues which needs to be addressed.

First, the use of dendrimer-based membranes for ion separation is not new - and citations to recent papers (e.g. Z-L, Qiu et al. ACS Nano 15, 7522-7535, 2021) are missing.

Second, for the modeling work: why was the B3LYP functional chosen? There are other DFT functionals each with advantages and disadvantages (for a nice review see N. Mardirossian & M. Head-Gordon. MOLECULAR PHYSICS, 2017, 115(19), 2315-2372).

Third, what role (if any) does electronic polarization of the constituent molecular structure play in the separation mechanism and how is this addressed in the MD simulations?

Finally, how does the MD PBC setup (which is indeed a good starting point) reflect the true corrugated membrane surface - and what implications does this simplification have for the results presented?

Response to Reviewer 1

Reviewer #1 (Remarks to the Author):

Comments to the Author

This study synthesized functionalized dendrimers (DA-G4D, BA-G4D, p-HC-G4D) with different functional groups that preferentially self-assembled in PIP solution to facilitate the formation of polyamide nanofilms with well-defined effective pore range. Compared with the pristine PA membrane, the self-assembled dendrimers (SADs) polyamide nanofilms achieved a high Cl⁻/SO₄²⁻ selectivity more than 17 times as compared with the pristine polyamide. The separation of Li⁺/Mg²⁺, achieving higher Li purity and recovery.

I do recommend publishing in Nature communication after major revision.

Response:

Thanks very much for your kind comments. We have carefully read your comments and gave the point-by-point response.

1: The separation factor in Supplementary Table 1 and Supplementary Table 4 is not consistent with the Supplementary Equation 3. In Equation 3 is α , but in the table is S. Please modified.

Response:

Thank you very much for your suggestion.

Based on your comment, the separation factor in Supplementary Tables 1–5 has been modified as " α ", which is consistent with the Supplementary Equation 3.

2: Why the model analysis pressure in Figure 4h is 4 bar, but 1 MPa is selected in experiment part.

After 70 % water recover, is the osmotic pressure of the salt solution higher than 4 bar?

Response:

Thank you very much for your suggestion.

Fig. 1 Module analysis: Calculated Li⁺ recovery vs water recovery along a simulated module. Feed conditions: Li⁺ concentration of 3.4 mM, Mg²⁺ of 19.6 mM, and operating pressure of 10 bar (1 MPa). Membrane properties: choosing the best-reported data for each type of membrane¹⁻¹².

Fig. 2 Module analysis: Calculated Li⁺ recovery vs membrane area along a

simulated module. Feed conditions: Li^+ concentration of 3.4 mM, Mg^{2+} of 19.6 mM, and operating pressure of 10 bar (1 MPa). Membrane properties: choosing the best-reported data for each type of membrane¹⁻¹².

(1) The reason that we chose 4 bar as the model analysis pressure in Figure 4h is because, some of literature reported membranes used to conduct the model analysis were tested at 4 bar, such as MBCN-0.02, PES/PSS/PAH(LbL), PSS/PAH2.5 (LbL), TFN/PEI-3 and Dual-skin layer NF (see Supplementary Table 6). Hence, 4 bar was selected as the model analysis pressure.

Besides, based on your comment, the model analysis was also carried out at the pressure of 10 bar (1 MPa). The new figure also was added in the Supplementary Information (see Supplementary Figure 38). As shown in Fig. 1, the designed membranes (A-PA/DA-G4D, A-PA/BA-G4D and A-PA/p-HC-G4D) in this paper shows higher performance than the membranes reported in literature. Specifically, when producing the same water recovery under operational pressure of 1 MPa, higher Li recovery is achieved with the A-PA/DA-G4D, A-PA/BA-G4D and A-PA/p-HC-G4D membranes using one element in series as compared with other membranes including commercial NF 90, Desal DL and other literature reported membranes. Moreover, it is noted that Desal DL appears to show a comparable performance at the lower water recovery (less than 25%). However, the water permeance of the Desal DL membrane (6.1 LMH/bar) are much less than that of the obtained membranes in this work (>20 LMH/bar). This implies that, compared to the resultant membranes of this work, much more membrane area is required for the Desal DL membrane to achieve the same Li recovery, as confirmed in Fig. 2. The feature of the obtained membranes in this work is high Li permeability at high water permeability. Therefore, the conclusion in our original manuscript remains unchanged and additional explanation has been added in the revised manuscript.

Fig. 3 Module analysis: Calculated osmotic pressure vs water recovery along a simulated module. Feed conditions: Li^+ concentration of 3.4 mM, Mg^{2+} of 19.6 mM, and operating pressure of 4 bar. Membrane properties: choosing the best-reported data for each type of membrane¹⁻¹².

Fig. 4 Module analysis: Calculated osmotic pressure vs water recovery along

a simulated module. Feed conditions: Li^+ concentration of 3.4 mM, Mg^{2+} of 19.6 mM, and operating pressure of 10 bar (1 MPa). Membrane properties: choosing the best-reported data for each type of membrane¹⁻¹².

(2) As demonstrated in Fig. 3, just as you point out, after 70% water recovery, the osmotic pressure approaches the operating pressure of 4 bar. Consequently, it becomes difficult to obtain convergence for the model solving process.

Moreover, with a higher pressure of 1 MPa (Fig. 4), the convergence problem becomes insignificant. As demonstrated in Fig. 4, the obtained model results for the full range of water recovery (0–100%) are displayed, basically showing no convergence problem during the model solving process. Specifically, although water recovery is as high as about 88%, model results can be obtained successfully. However, during the practical application of Li recovery, water recovery of higher than 90% seems to be meaningless.

Reference

1. Pramanik, B. K., Asif, M. B., Kentish, S., Nghiem, L. D. & Hai, F. I. Lithium enrichment from a simulated salt lake brine using an integrated nanofiltration-membrane distillation process. *J. Environ. Chem. Eng.* **7**, 103395 (2019).
2. Li, X., Zhang, C., Zhang, S., Li, J., He, B. & Cui, Z. Preparation and characterization of positively charged polyamide composite nanofiltration hollow fiber membrane for lithium and magnesium separation. *Desalination* **369**, 26–36 (2015).
3. Sun, S-Y., Cai, L-J., Nie, X-Y., Song, X. & Yu, J-G. Separation of magnesium and lithium from brine using a Desal nanofiltration membrane. *J. Water Process Eng.* **7**, 210–217 (2015).
4. Yang, Z., Fang, W., Wang, Z., Zhang, R., Zhu, Y. & Jin, J. Dual-skin layer nanofiltration membranes for highly selective $\text{Li}^+/\text{Mg}^{2+}$ separation. *J. Memb. Sci.* **620**, 118862 (2021).
5. He, R., Dong, C., Xu, S., Liu, C., Zhao, S. & He, T. Unprecedented $\text{Mg}^{2+}/\text{Li}^+$ separation using layer-by-layer based nanofiltration hollow fiber membranes. *Desalination* **525**, 115492 (2022).
6. Bi, Q., Zhang, C., Liu, J., Liu, X. & Xu, S. Positively charged zwitterion-carbon nitride functionalized nanofiltration membranes with excellent separation performance of $\text{Mg}^{2+}/\text{Li}^+$ and good antifouling properties. *Sep. Purif. Technol.* **257**, 117959 (2021).
7. Guo, C. et al. Ultra-thin double Janus nanofiltration membrane for

separation of Li⁺ and Mg²⁺: “Drag” effect from carboxyl-containing negative interlayer. *Sep. Purif. Technol.* **230**, 115567 (2020).

8. Lu, D. et al. Constructing a selective blocked-nanolayer on nanofiltration membrane via surface-charge inversion for promoting Li⁺ permselectivity over Mg²⁺. *J. Memb. Sci.* **635**, 119504 (2021).
9. Aghili, F., Ghoreyshi, A. A., Van der Bruggen, B. & Rahimpour, A. A highly permeable UiO-66-NH₂/polyethyleneimine thin-film nanocomposite membrane for recovery of valuable metal ions from brackish water. *Process Saf. Environ.* **151**, 244–256 (2021).
10. Wang, L. et al. Novel positively charged metal-coordinated nanofiltration membrane for lithium recovery. *ACS Appl. Mater. Inter.* **13**, 16906–16915 (2021).
11. Li, W., Shi, C., Zhou, A., He, X., Sun, Y. & Zhang, J. A positively charged composite nanofiltration membrane modified by EDTA for LiCl/MgCl₂ separation. *Sep. Purif. Technol.* **186**, 233–242 (2017).
12. He, R., Xu, S., Wang, R., Bai, B., Lin, S. & He, T. Polyelectrolyte-based nanofiltration membranes with exceptional performance in Mg²⁺/Li⁺ separation in a wide range of solution conditions. *J. Memb. Sci.* **663**, 121027 (2022).

3: Page 15, Line 335-338. Why is Li⁺ ion negatively intercepted? There is no explanation here.

Response:

Thank you very much for your suggestion.

Below is the explanation that why Li⁺ ion negatively intercepted. Please see Line 424 in the marked version of manuscript.

In regard to the negative rejection rate of Li⁺, the ion rejection of mixed solution is generally impacted by the concentration of other ions and their difference among materialized parameters. As illustrated in Supplementary Table 4, the Li⁺ concentration in the LiCl/MgCl₂ solution is far less than the Cl⁻. Take 31.2 of MLR (containing 0.5 g L⁻¹ LiCl and 10 g L⁻¹ MgCl₂) as an example, the total Cl⁻ and Mg²⁺ concentration was 7889.3 ppm and 2552.75 ppm, whereas the Li⁺ was only 81.87 ppm. When ions permeate through the PA membrane, there was always residual Mg²⁺ that has not entirely been intercepted in permeation solution, then make it possible that the Cl⁻

concentration of the permeation side was larger than that of initial 418.14 ppm from 0.5 g L⁻¹ LiCl. Therefore, due to the Donnan equilibrium effect, Li⁺ concentration in permeation side was more likely to even more than the feed side. In addition, the Li⁺ has a more significant diffusion coefficient ($1.03 \times 10^{-9} \text{ m}^2 \text{ s}^{-1}$) and smaller hydrated radius (0.382 nm) than Mg²⁺ ($0.706 \times 10^{-9} \text{ m}^2 \text{ s}^{-1}$, 0.428 nm), which is easier to pass through the nanofilm (Supplementary Table 8). It is thus estimated that the retardance of Mg²⁺ also results in the negative rejection rate of Li⁺. Furthermore, the hydration energy of Li⁺ (474 kJ mol⁻¹) is lower than Mg²⁺ (1828 kJ mol⁻¹) (Supplementary Table 8), which also is helpful to easily transport through the PA nanofilm¹. The previously reported literatures also have been confirmed that the negative rejection rate of Li⁺ is rational and beneficial, enabling the enhancement of the separation factor of Li⁺/Mg²⁺²⁻⁴.

Reference

1. Xu, S. et al. Extraction of lithium from Chinese salt-lake brines by membranes: Design and practice, *J. Membr. Sci.* **635**, 119441 (2021).
2. Li, Y., Zhao, Y., Wang, H. & Wang, M. The application of nanofiltration membrane for recovering lithium from salt lake brine, *Desalination* **468**, 114081 (2019).
3. Guo, Y., Ying, Y., Mao, Y., Peng, X. & Chen, B. Polystyrene sulfonate threaded through a metal-organic framework membrane for fast and selective lithium-ion separation. *Angew. Chem. Int. Edit.* **55**, 15120–15124 (2016).
4. He, R., Dong, C., Xu, S., Liu, C., Zhao, S. He, T. Unprecedented Mg²⁺/Li⁺ separation using layer-by-layer based nanofiltration hollow fiber membranes. *Desalination* **525**, 115492 (2022).

4: Page 9, Line 3197-199: 'The upper surfaces images from SEM showed the existence of SADs (DA-G4D, BA-G4D and p-HC-G4D) nanoparticles covalently linked within the surrounding of the nanostripe PA layer (Figs. 2b-2d, Supplementary Figs. 18-20).' It is difficult to judge covalently linked from SEM. Please explain.

Response:

Thank you very much for your suggestion.

Based on your comment, explanation and modification on this issue are as

follows.

According to the zeta potential value of the self-assembled dendrimers (SADs), under the optimized PIP (0.8 wt.%) and dendrimer (0.04 wt.% or 0.08 wt.%) concentrations for the preparation of PA nanofilms, the formed SADs exhibit a positive charge, demonstrating that the periphery of the SADs possess lots of PIP molecules. In other words, during interfacial polymerization process, the PIP molecules around the SADs can be involved in the interfacial reaction and form the covalent bond. This process imparts a good compatibility between SADs and polyamide segments.

Hence, the related expression has been modified as the follows. Please see Line 234 in the marked version of manuscript.

"The upper surfaces images from SEM (Figs. 2b–2d, Supplementary Figs. 21–23) showed the existence of SADs (DA-G4D, BA-G4D and p-HC-G4D) nanoparticles that were embedded within the surrounding of the nanostripe PA layer, demonstrating a good compatibility. This stemmed from that there were aggregative PIP molecules on the periphery of the SADs that can be involved in the interfacial reaction."

5: ① Page 7, Line 168-170: 'The BA-G4D and p-HC-G4D SADs exhibit a negative surface charge under a 0.01 wt.% PIP and 0.01 wt.% dendrimer (Supplementary Figs. 16a–16c).'

However, the concentrations in Supplementary Figs. 16a–16c were not 0.01 wt.% PIP and 0.01 wt.% dendrimer.

② Page 7, Line 170-174: 'With the increased concentration (0.8 wt.% PIP and 0.04 wt.% dendrimer), the DA-G4D, BA-G4D and p-HC-G4D SADs possess a positive surface charge, which indicates that the peripheries of SADs are covered with PIP molecules and can be involved in the IP to form amide bond, and fine-tune the nanofilm inner structure (Supplementary Figs. 16d–16f).' But this sentence is not a description of Supplementary Figs. 16d–16f.

Please modify the description of Supplementary Figs. 16.

Response:

Thank you very much for your suggestion.

Based on your comment, the description of Supplementary Figs. 16 (now Supplementary Figs. 19 in Supplementary Information) in manuscript has been modified and corrected. Please refer to Line 183–189 in the marked version of

manuscript.

The details are as follows.

"The BA-G4D and p-HC-G4D SADs exhibit a negative surface charge under a 0.01 wt.% PIP and 0.01 wt.% dendrimer (Supplementary Figs. 19a–19c). With the increase of PIP and dendrimer concentrations (0.8 wt.% PIP and 0.04 wt.% dendrimer), the DA-G4D, BA-G4D and p-HC-G4D SADs possess a positive surface charge, which indicates that the peripheries of SADs are featured with aggregated PIP molecules and can be involved in the IP to form amide bond, and fine-tune the nanofilm inner structure (Supplementary Figs. 19d–19f)."

6: What is the concentration of PIP and synthesized functionalized dendrimers (DA-G4D, BA-G4D, p-HC-G4D) in Figure 3 and Figure 2?

Response:

Thank you very much for your suggestion.

The concentrations of the PIP and DA-G4D, BA-G4D and p-HC-G4D dendrimers in Figure 3 and Figure 2 were 0.8 wt.%, 0.04 wt.%, 0.04 wt.% and 0.08 wt.%, respectively.

7. In Figure 1: What state was the SADs in PIP water phase? In single particle or aggregated state.

Response:

Thank you very much for your suggestion.

As shown in Supplementary Figures 9–14, based on the HRTEM and SEM images, the SADs in PIP water phase are more likely to show the single particle state.

8. The SADs (DA-G4D, BA-G4D and p-HC-G4D) have dendrimers structure. The structure shows large voids. What is the diameter of the voids? These voids allow passage of water. Do they also allow passage of ions?

Response:

Thank you very much for your suggestion.

As shown in Fig.1, according to the high-resolution transmission electron microscopy (HRTEM) images of the SADs (DA-G4D, BA-G4D and p-HC-G4D), we measured that the diameter of the nanovoids in these SADs were ranged from 0.96 nm to 1.12 nm. Hence, based on the hydrated radius of the Mg^{2+} (0.428 nm), Cl^- (0.332 nm), SO_4^{2-} (0.379 nm) and Li^+ (0.382 nm), these voids also allowed the passage of ions.

The related phrase has been added in the Line 149–153 of the marked version of manuscript.

Fig. 1 High-resolution transmission electron microscopy (HRTEM) images of the SADs (DA-G4D, BA-G4D and p-HC-G4D). The white circles in the picture represent nanovoids.

9. How to calculate the volumetric charge densities according to zeta potential values in Figure 3a?

Response:

Thank you very much for your suggestion.

Based on your comment, the following is the Equations for calculating the volumetric charge densities. Please refer to the section 2.7 of Supplementary

Information.

According to the method reported by Shardul S. Wadekar and Radisav D. Vidic¹, using Equations (1) and (2), we calculated the σ_{ek} , electrokinetic charge density (C/m²), and $-X$, volumetric charge densities (mol m⁻³):

$$\sigma_{ek} = -\text{sign}(\zeta) \sqrt{\left(2\epsilon_0\epsilon_b RT \sum_i c_i^b \left[\exp\left(\frac{-z_i F \zeta}{RT}\right) - 1\right]\right)} \quad (1)$$

$$-X = \frac{\sigma_{ek}}{\gamma_p F} \quad (2)$$

Of which, the γ_p is the same as the μ_p (mean effective pore radius), ϵ_0 is vacuum permittivity (8.854×10⁻¹² C/(mV)), ϵ_b is dielectric constant for the feed salt solution (assumed to be equal to that of water at 25°C=80.1), R is universal gas constant (8.31446 J/(mol·K)), T is absolute temperature (296.15 K), c_i^b is bulk feed concentration of ion i, z_i is valence of ion i, F is Faraday constant (96485.3329 C/mol).

Reference

1. Wadekar, S. S. & Vidic, R. D. Insights into the rejection of barium and strontium by nanofiltration membrane from experimental and modeling analysis. *J. Membr. Sci.* **564**, 742–752 (2018).

10. Figure 5 c, d, e does not correspond to the legend description.

Response:

Thank you very much for your suggestion.

Based on your comments, the legend descriptions of Figures 5 c, 5d and 5e have been corrected. Please refer to Line 501–503 in the marked version of manuscript.

Response to Reviewer 2

Reviewer #2 (Remarks to the Author):

Recommendation: Reconsider after major changes Yuan et al. reported the synthesis of polyamide membranes modified with dendrimers and their use in the separation of ions of similar sizes. In their investigation, the authors hypothesized that dendrimers could be used to modify the pore volume of polyamide membranes to make them more selective for ion separation. The authors synthesized an amine-terminal dendrimer and functionalized it to make it water-soluble. The authors fabricated membranes using dendrimers and polyamides on a polysulfone support, which were characterized and used in ion rejection experiments. The authors performed a series of comparisons with a control membrane and commercial membranes to demonstrate the superior performance of dendrimer/polyamide membranes. Simulations were performed to support the differential ion rejection findings.

The following general points must be addressed for the manuscript to be considered for publication.

1. The manuscript requires grammar improvement to deliver a clearer message to the audience. Some areas of improvement are addressed in the specific comments.

Response:

Thanks very much for your kind suggestion.

According to your comments, the grammar in manuscript has been carefully checked and corrected to deliver clearly to the audience. Your suggestions certainly make this manuscript clear and intelligible.

Please refer to the marked version of manuscript.

2. The manuscript requires a better organization of the figures, especially in the supporting information. The authors need to consider a reorganization that makes the text reading more congruent and organized with the figures presented.

Response:

Thanks very much for your kind suggestion.

The figures in manuscript and supplementary information have been reorganized. And the text to read is more congruent and organized under your comments.

Please refer to the marked version of manuscript.

3. This manuscript has many claims that are not supported by the results. Authors must remove some of the claims or perform experiments that support them.

Response:

Thanks very much for your kind suggestion.

According to your comments, the claims that are not supported by the results have given the related explanations and removed.

Please refer to the marked version of manuscript.

4. Although the authors tested three different membranes, there is no clear rationale for why one is better than the others. The authors need to include an analysis that correlates the properties of the different dendrimers with the rejection study results.

Response:

Thanks very much for your kind suggestion.

According to your comments, the related analyses and discussion that why one is better than the others have been added. Please refer to the response to related comment, for example 23.

The following are specific comments:

1. Line 45-47: Consider rephrasing by combining the two phrases to form a sentence.

Response:

Thank you very much for your suggestion.

Based on your comments, the two phrases have been formed a sentence as follows: "The superior permeabilities were maintained by forming the asymmetric hollow nanostripe nanofilms, and their well-designed ion separation pore range showed an enhancement, rationalized by molecular simulation." Please see Line 42–45 in the marked version of manuscript.

2. Line 50-51: Hard to follow and consider rephrasing.

Response:

Thank you very much for your suggestion.

Based on your comments, the related sentence has been corrected as the follows, "Furthermore, achieving higher Li purity and Li recovery was observed for the designed membranes, better than current state-of-the-art membranes." Please see Line 47–49 in the marked version of manuscript.

3. Line 133-135: Do the authors have DLS data to support the HRTEM findings?

Response:

Thank you very much for your suggestion.

As you comment, we conducted the DLS measurement, and the DLS data of the single dendrimer nanoparticle sizes were shown in Fig. 1 (Supplementary Fig. 17d).

Fig. 1 Nanoparticle size distribution of the self-assembled dendrimers (SADs) in 0.8 wt.% PIP and 1 wt.% dendrimer concentration when diluted 20 times with 0.1 wt.% PIP concentration. The final concentrations calculated for PIP and dendrimer are 0.135 wt.% and 0.05 wt.%, respectively.

4. Line 139: What image?

Response:

Thank you very much for your suggestion.

The image we mentioned here is the electronic photos of the transparent sample bottle containing self-assembled dendrimers (SADs) solution in Figs. 1b, 1c and 1d.

Based on your comments, the related sentence has been corrected as follows, "The images of the transparent sample bottle containing solution in Figs. 1b, 1c and 1d showed electronic photos of the SADs aqueous-phase PIP solution after stable storage for one year." Please see Line 137–139 in the marked version of manuscript.

5. Line 140-142: Consider rephrasing.

Response:

Thank you very much for your suggestion.

Based on your comments, the related sentence has been corrected as the follows, "More importantly, self-assemble behavior was found when DA-G4D, BA-G4D and p-HC-G4D dendrimers were dissolved in PIP solution." Please see Line 139–141 in the marked version of manuscript.

6. Line 156-159: Any suggestion why DA-G4D behaves differently?

Response:

Thank you very much for your suggestion.

Based on your comment, as shown in Fig. 1, we investigated the nanoparticle size distribution of the DA-G4D self-assembled dendrimers (SADs) in different PIP concentrations. We estimated that DA-G4D behaves differently was because that it has the aliphatic chain carboxyl group, which was more likely to move in aqueous solution rather than anchor by electrostatic interaction as compared with aromatic carboxyl (BA-G4D) and phenolic hydroxyl groups (p-HC-G4D). In other words, the DA-G4D dendrimers with aliphatic chain carboxyl groups needed more PIP molecules to generate electrostatic interaction and form the large-sized self-assembled dendrimers. This conclusion has been confirmed in Fig. 1 and Supplementary Fig. 17a (Supplementary Information). As demonstrated in Fig. 1, when the PIP concentration was increased up to 0.02wt.%, 0.03 wt.% and 0.05 wt.%, sizes of the DA-G4D self-assembled dendrimers (SADs) were turned into large accordingly.

The related explanation has been added in Line 160–169 in the marked version of manuscript.

Fig. 1 Nanoparticle size distribution of the DA-G4D self-assembled dendrimers (SADs) in different PIP concentrations.

7. Line 159-161: There is a mismatch between the figures and the concentrations.

Response:

Thank you very much for your suggestion.

Based on your comments, the figures and the concentrations have been corrected and matched. Please see Line 169–171 in the marked version of manuscript.

8. Line 161-163: What is the final concentration of dendrimers and PIP? Why are all dendrimers single nanoparticles?

Fig. 1 Size distribution (a, b, c) and zeta potentials (d, e, f) of the self-assemble dendrimers (composition: 0.8 wt.% PIP and 1 wt.% dendrimer) before and after dilution using 0.1 wt.% PIP concentration.

Response:

Thank you very much for your suggestion.

The final concentrations calculated for PIP and dendrimer are 0.135 wt.% and 0.05 wt.%, respectively.

Below are the explanations that why are all dendrimers single nanoparticles after dilution. Please see Line 173–179 in the marked version of manuscript.

Based on your comment, we investigated the size distribution and zeta potentials of the self-assemble dendrimers before and after dilution using 0.1 wt.% PIP concentration. As shown in Figs. 1a-1c, when using 0.1 wt.% PIP to dilute the 0.8 wt.% and 1 wt.% dendrimer (SADs), these self-assemble dendrimers (SADs) became the single dendrimer nanoparticle, which was consistent with our previous results. Zeta potentials data in Figs. 1d-1f further illustrated that, these self-assemble dendrimers (SADs) before dilution basically all showed a neutral charge surface, while after dilution, their surfaces turned

into negative charges. These results demonstrated that using 0.1 wt.% PIP to dilute the SADs solution (containing 0.8 wt.% PIP and 1 wt.% dendrimer) can alter the electrostatic interaction between dendrimers and PIP molecules, further resulted in the disaggregation of the SADs formed, then finally turned into the single dendrimer nanoparticle.

9. Line 174-177: Which ones are the best conditions? What are the criteria for their selection?

Response:

Thank you very much for your suggestion.

Based on your comment, the best conditions for the polyamide nanofilm used for ion separation are as follows. For the A-PA/DA-G4D, its best condition was 0.8 wt.% PIP and 0.04 wt.% DA-G4D. The best condition for the A-PA/BA-G4D was 0.8 wt.% PIP and 0.04 wt.% BA-G4D, while that of the A-PA/p-HC-G4D was 0.8 wt.% PIP and 0.08 wt.% BA-G4D.

In fact, there were several criteria for the selection in this work, one was the direct desalination or ion separation performance of the resulted membranes, another was the zeta potential or morphology structure of self-assemble dendrimers (SADs). Positive of the zeta potential value ensures that there was sufficient PIP molecule on periphery of SADs, which can participate in the IP process, leading to the enhancement of interface compatibility and stability of imbedding.

The related phrase has been added at Line 192–201 in the marked version of manuscript.

10. Line 181-183: I do not follow the discussion; what does the authors mean by the inconsonant diffusion-reaction rate?

Response:

Thank you very much for your suggestion.

Based on your comment, the discussion has been modified as the follows.

It is known that, the PSF support characteristic such as pore size, uniformity, wettability, and morphology, can affect the amine molecule distribution, further vary its diffusion-reaction rate, and then result in the formation of a

heterogeneous polyamide nanofilm during interfacial polymerization¹⁻³. Previous reports demonstrated that after formation of dendrimer porous layer on PSF support, compared with the pristine PSF support, polar part of solid surface energy was increased, while that of the non-polar part was decreased and its pore size was also decreased⁴. When the aqueous PIP solution was soaked into the PSF support with dendrimer porous layer, it interacted with the dendrimer porous layer via hydrogen bonding and led to an inhomogeneous PIP solutions distribution, reducing the diffusion rate of the PIP molecule⁵. Hence, diffusion-reaction behaviors of PIP molecules from aqueous phase to organic phase that conducted on the modified PSF support with dendrimer porous layer was different with that of the pristine PSF support. It can be rationally speculated that such diffusion-reaction was inconsonant or non-uniform at nanoscale, where some IP reaction sites dominated by the dendrimer porous layer, whereas other sites still affected by the PSF support. This diffusion-driven instability facilitates the formation of nano-stripe structure⁵.

Please refer to the Line 205–217 in the marked version of manuscript.

Reference

1. Ghosh, A. K. & Hoek, E. M. V. Impacts of support membrane structure and chemistry on polyamide-polysulfone interfacial composite membranes. *J. Membr. Sci.* **336**, 140–148 (2009).
2. Morgan, P. W. & Kwolek, S. L. Interfacial polycondensation. II. Fundamentals of polymer formation at liquid interfaces. *J. Polym. Sci. Pol. Chem.* **40**, 299–327 (1959).
3. Morgan, P. W. & Kwolek, S. L. Interfacial polycondensation. II. Fundamentals of polymer formation at liquid interfaces. *J. Polym. Sci. Pol. Chem.* **34**, 531–559 (1996).
4. Yuan, B., Zhao, S., Hu, P., Cui, J., & Niu, Q. J. Asymmetric polyamide 634 nanofilms with highly ordered nanovoids for water purification. *Nat. Commun.* **11**, 6102 (2020).
5. Tan, Z., Chen, S., Peng, X., Zhang, L. & Gao, C. Polyamide membranes with nanoscale Turing structures for water purification. *Science* **360**, 518–521 (2018).

11. Line 183-186: It is not clear how the polyamide film is formed. What is the amount of PIP required to fabricate the polyamide film? For the control (A-PA)

how much PIP was used? Do the authors use another diamine to form the polyamide films?

Response:

Thank you very much for your suggestion.

According to your comment, the amount of PIP required to fabricate the polyamide nanofilm was 0.8 wt.%. The control (A-PA) polyamide nanofilm for the PIP also was 0.8 wt.%. In this work, we did not use another diamine to form the polyamide nanofilms.

The sentence here refers that, dendrimer porous layer onto the PSF support possessed residual aromatic amine functional groups after diazotization-coupling reaction. Hence, these remained amine groups in the dendrimer porous layer can participate in the IP reaction with TMC, forming a polyamide nanofilm with a two-layer nanostructure via stable covalent bonds.

The related sentences have been modified, please see Line 217–220 in the marked version of manuscript.

12. Line 188-190: The authors claim a covalent bond between the SADs and the PA film. However, no experiment has supported this claim. Is the bond between the dendrimers and PIP covalent? How do they know that PIP is covalently attached to PA films?

Response:

Thank you very much for your suggestion.

As you comment, there is indeed no covalent bond between dendrimers and PIP or PA layer.

Based on the zeta potential value of the SADs PIP solution, it showed positive charge, indicating that periphery of SADs was attached with PIP molecules (Supplementary Figs 19d–19f). Hence, as illustrated in Fig. 1, we estimated that these peripheral PIP molecules in the SADs solution can react with TMC during IP process to form amide bond. After completion of IP, these SADs were just imbedded into the PA layer, showing a good compatibility and stability.

Here, we suggest that, under the electrostatic self-assembly mechanism of aqueous solution state, since SADs has peripheral PIP molecules, using the “covalently linked” to describe the IP reaction stage between SADs PIP solution

and TMC is applicable. However, once IP completed and PA nanofilm formed, these SADs were imbedded into PA nanofilm, thus there were no covalent bond.

Hence, based on your comment, the claim on a covalent bond between the SADs and the PA film has been deleted and modified as follows.

“With the formation of SADs aqueous-phase PIP solution, when applied in the IP stage, peripheral PIP molecules of the SADs can react with TMC during IP process to form amide bond, leading to the enhancement of their interface compatibility and stability of imbedding.”

Please see Line 222–225 in the marked version of manuscript.

Fig. 1 Schematic diagram of individual BA-G4D SADs nanoparticles in self-assembled dendrimers (SADs) aqueous-phase PIP solutions.

13. Line 1998: SEM cannot be used to claim a covalent linkage between the dendrimers and the PA film.

Response:

Thank you very much for your suggestion.

Based on your comment, the claim on a covalent linkage between the dendrimers and the PA film has been deleted and modified as follows.

“The upper surfaces images from SEM (Figs. 2b–2d, Supplementary Figs. 21–23) showed the existence of SADs (DA-G4D, BA-G4D and p-HC-G4D) nanoparticles that were embedded within the surrounding of the nanostripe PA layer, demonstrating a good compatibility.”

Please see Line 234–237 in the marked version of manuscript.

14. Line 213-218: It is not clear to me what is the relationship between layer thickness and peripheral PIP molecules. If dendrimers with a lower amount of PIP were to be used, the thickness of the PA film would be higher? What are the errors in thickness values?

Fig. 1 Cross-sectional morphology of SADs polyamide nanofilms characterized by SEM (a, A-PA, b, A-PA/DA-G4D, c, A-PA/BA-G4D, and d, A-PA/p-HC-G4D). Fabrication condition: A-PA (0.6 wt.% PIP + 0.1 w/v% TMC, 30s), A-PA/DA-G4D (0.6 wt.% PIP+0.04 wt.%DA-G4D + 0.1 w/v% TMC, 30s), A-PA/BA-G4D (0.6 wt.% PIP+0.04 wt.% BA-G4D + 0.1 w/v% TMC, 30s) and A-PA/p-HC-G4D (0.6 wt.% PIP+0.08 wt.% p-HC-G4D + 0.1 w/v% TMC, 30s).

Fig. 2 a Fabrication schematic of SADs polyamide nanofilm with nanostripe asymmetric structure by IP at an interface between TMC solution in cyclohexane and an SADs PIP aqueous solution. **b** Fabrication schematic of pristine polyamide nanofilm with nanostripe asymmetric structure by IP at an interface between TMC solution in cyclohexane and PIP aqueous solution.

Response:

Thank you very much for your suggestion.

Based on your comment, we investigated the influence of dendrimers with a lower amount of PIP to react with TMC on thickness of the polyamide nanofilm, specifically, using a 0.6 wt.% PIP. As shown in Fig. 1, the thicknesses of the polyamide nanofilms using a 0.6 wt.% PIP was lower than these polyamide nanofilms made by 0.8 wt.% PIP (see Fig. 2 in Manuscript). For example, when using 0.6 wt.% PIP, thickness of the A-PA/p-HC-G4D dense layer is 28 nm, while it became 51.4 nm after using 0.8 wt.% PIP (see Fig. 2 in Manuscript). Moreover, as demonstrated in Fig. 2a (see manuscript that using 0.8 wt.% PIP)

and Fig. 1a (using 0.6 wt.% PIP), the thickness for the dense layer (upper layer) observed with the incorporation of self-assemble dendrimers (SADs) is enhanced as compared with those of the pristine A-PA nanofilm.

As for the errors in thickness values, based on the above discussions, we rationally speculated that there are two main reasons.

As illustrated in Fig.2, one is because that the self-assembled dendrimers (SADs), featured by aggregated PIP molecules on their peripheries, can be located at the interface of the aqueous and organic phase during IP process, then indirectly increase the PIP concentration of interface to facilitate the formation of a thicker thickness of polyamide nanofilm.

Another is that these self-assembled dendrimers (SADs), with intrinsic nanoscale size and aggregated PIP molecules, can participate in the IP (see Supplementary Figs. 17–19) and be embedded into the polyamide nanofilm, then increase the thickness of polyamide nanofilm.

The related explanation has been added in marked version of manuscript, please see Line 255–262.

15. Line 218-221: Consider rephrasing.

Response:

Thank you very much for your suggestion.

Based on your comments, the sentence in Line 218–221 has been corrected as follows,

“Moreover, for cross-section images of TEM, since the resin used by the slicing was readily permeated into the dendrimer porous layer, like permeating the PSF layer with inner loose structure. The sublayer (dendrimer porous layer) thus cannot be observed in TEM imaging as like the cross-section images of SEM.”

Please see Line 263–266 in marked version of manuscript.

16. Line 221: Why is it clear that nanovoids are present?

Response:

Thank you very much for your suggestion.

The meaning of this sentence is that, the results of cross-sectional

morphology of nanofilms characterized by TEM are matched with the bottom to top micrographs of nanofilms obtained from TEM and SEM, where both showed the nanovoids. Hence, we consider that the present of nanovoids is clear and sounded, and this is also confirmed by other literature reports.

For clearer expression, the related sentence has been modified as follows,

“Besides, nanovoids consisting of nanostripes were clearly observed in the cross-sectional morphology of nanofilms characterized by TEM and the bottom to top micrographs of nanofilms obtained from TEM and SEM, where such morphology also contributed to provide a higher water permeation rate.”

Please see Line 266–270 in marked version of manuscript.

17. Line 223-227: Authors claim that dendrimers have a negative charge. This claim seems to be true when 0.01% SADs concentration was used. However, when 0.04 and 0.08 % SADs were used, the charge was positive. In the methods section (line 508), the authors stated that 0.04 % and 0.08 % SADs concentrations were used. Why the discrepancy?

Response:

Thank you very much for your suggestion.

As illustrated in manuscript, the designed DA-G4D, BA-G4D and p-HC-G4D dendrimers possess carboxylic acid group and phenolic hydroxyl group, respectively. These dendrimers can deprotonate in aqueous PIP solution and interact with PIP molecule, forming self-assembled dendrimers (SADs). When 0.01% dendrimers and 0.01% PIP concentrations were used, the periphery of the formed SADs (0.01% SADs concentration) has not sufficient PIP molecule, then shows a negative charge. Further, when 0.04%/0.08% dendrimers and 0.8% PIP concentrations were used, the periphery of the formed SADs (0.04%/0.08% SADs concentration) has sufficient PIP molecule, thus exhibits a positive charge.

As illustrated in the response to comment 9, considering the ion separation performance and interface compatibility of the SADs in the fabricated of polyamide nanofilms, we explored the 0.04% and 0.08% SADs concentrations as the best conditions.

In fact, since there is sufficient PIP molecule on periphery of SADs, these peripheral PIP molecules can participate in the interfacial reaction to form amide bonds. After that, these SADs were imbedded into the polyamide

nanofilm with good interface compatibility and stability. Hence, because the peripheral PIP molecules on the SADs were used to form amide bonds, the SADs in polyamide nanofilm show a negative charge due to the feature of the carboxylic acid group and phenolic hydroxyl group.

18. Line 224: How do the authors know that there is a covalent bond?

Response:

Thank you very much for your suggestion.

Based on your comment, the claim on a covalent linkage between the dendrimers and the PA film has been deleted. Please refer to Line 272 in marked version of manuscript.

19. Line 247-248: What is the importance of membranes with similar wettability?

Response:

Thank you very much for your suggestion.

It is known that wettability is one of the key factors that affect the water/salt permeance flux. Hence, the designed polyamide membranes and the controlled membrane with similar wettability were observed, it means that, when we studied and discussed the difference of the resultant membranes in terms of water/salt permeance flux, we would no longer take into account the wettability as a factor to influence flux. We will consider the influence of the resultant membranes in terms of thickness and surface morphology on the water/salt permeance flux.

The related explanation has been added in Line 297–301 of the marked version of manuscript.

20. Line 267-271; Consider discussing the water flux for LiCl as well.

Response:

Thank you very much for your suggestion.

Based on your comment, the discussing about the water flux for LiCl was

added in manuscript, and below is the detail.

“Correspondingly, the water flux for LiCl showed a reduction from 302.32 to 223.53 L m⁻² h⁻¹.”

Please see Line 337 in marked version of manuscript.

21. Line 278-279: Authors mention optimized conditions, but do not comment what are those conditions and how they obtained them.

Response:

Thank you very much for your suggestion.

Based on your comment, below are the optimized conditions to fabricate polyamide nanofilm used for ion separation.

For the A-PA/DA-G4D, its optimized condition was 0.8 wt.% PIP and 0.04 wt.% DA-G4D. The optimized condition for the A-PA/BA-G4D was 0.8 wt.% PIP and 0.04 wt.% BA-G4D, while that of the A-PA/p-HC-G4D was 0.8 wt.% PIP and 0.08 wt.% BA-G4D.

The processes to obtain the optimized conditions were as follows.

After explored the conditions used for the fabrication of the controlled polyamide nanofilm, we fixed the related PIP and TMC concentrations were 0.8 wt.% and 0.1 w/v%, respectively. Then, different dendrimer concentrations ranged from 0.02 wt.%, 0.04 wt.%, 0.06 wt.%, 0.08 wt.% and 0.1 wt.% were added into the 0.8 wt.% PIP to generate self-assembled dendrimers reactive solution, then to react with TMC to form SADs polyamide nanofilms. Through conducting the single salt separation performance tests, analysis and comparison, we defined the optimized conditions to fabricate the SADs polyamide nanofilms.

Please see Line 319–329 in marked version of manuscript.

22. Line 308-309: Authors should provide a context for why the studied ions are important.

Response:

Thank you very much for your suggestion.

Based on your comment, below is the context for why separation of Li⁺/Mg²⁺ and Cl⁻/SO₄²⁻ are important.

“Separation of Li⁺/Mg²⁺ and Cl⁻/SO₄²⁻ are considered as a significant

approach of achieving circularity in resource management, such as Salt Lake lithium extraction and industrial wastewater minimal or zero-liquid discharge strategy¹.”

Please see Line 373–375 in marked version of manuscript.

Reference

1. Wang, Z., Deshmukh, A., Du, Y. & Elimelech, M. Minimal and zero liquid discharge with reverse osmosis using low-salt-rejection membranes. *Water Res.* **170**, 115317 (2020).

23. Line 319-321: What is the rationale that makes A-PA/P-HC-G4D and A-PA/BA-G4D better than A-PA/DA-G4D?

Response:

Thank you very much for your suggestion.

Below are the explanations on the related statement.

It is known that the separation mechanism for nanofiltration membrane mainly includes size sieving and Donnan effect. As demonstrated in Supplementary Figure 32, the surface negative charge at pH=7 of the A-PA/BA-G4D (-42.26 mV) was higher than that of the A-PA/p-HC-G4D (-41.73 mV) and the A-PA/DA-G4D (-42.17 mV). Meanwhile, as shown in Fig. 5b of Manuscript, the Cl⁻/SO₄²⁻ separation area for the PA/p-HC-G4D (0.33) was larger compared with that of the PA/DA-G4D (0.28) and the PA/BA-G4D (0.16). Though the A-PA/DA-G4D exhibited a larger Cl⁻/SO₄²⁻ separation area than the A-PA/BA-G4D, the A-PA/BA-G4D has more Donnan effect on SO₄²⁻ than the A-PA/DA-G4D, further readily producing a higher SO₄²⁻ or Na₂SO₄ rejection. In other words, when separating Cl⁻/SO₄²⁻, the synergetic effects of size sieving and Donnan effect for the A-PA/p-HC-G4D and the A-PA/BA-G4D was better than that of the A-PA/DA-G4D. Hence, based on the equation 3 in Supplementary Information, the calculated Cl⁻/SO₄²⁻ separation factor for the A-PA/BA-G4D is better than that of the A-PA/DA-G4D. Similarly, due to the larger Cl⁻/SO₄²⁻ separation area for the A-PA/p-HC-G4D, its size sieving effect on separation of Cl⁻/SO₄²⁻ was more significant than that of the Donnan effect on separation of Cl⁻/SO₄²⁻ for the A-PA/DA-G4D. The A-PA/p-HC-G4D showed a higher Cl⁻/SO₄²⁻ separation factor compared with that of the A-PA/DA-G4D.

On the whole, since the difference of pore structure and inner charge for the

A-PA/P-HC-G4D, the A-PA/BA-G4D and the A-PA/DA-G4D, the separation roles including size sieving and Donnan effect of these designed polyamide nanofilms on $\text{Cl}^-/\text{SO}_4^{2-}$ showed corresponding strength difference. Based on the above discussions, these strength difference among separation roles make the A-PA/P-HC-G4D and the A-PA/BA-G4D better than the A-PA/DA-G4D.

The related explanation has been added in Line 388–397 of the marked version of manuscript.

24. Line 322-325: The authors used the same concentration for both Li and Mg for the high concentration studies. A more relevant investigation would be to use the average concentrations of these ions in salt-lake brines. How common are the selected concentrations used in this section?

Response:

Thank you very much for your suggestion.

Below is the explanation that how common are the selected concentrations used in this section.

Table 1 shows the Li^+ and Mg^{2+} mass concentration (g L^{-1}) used in this section. Table 2 presents the Li^+ and Mg^{2+} mass concentration of various brine of commercial value around the world. Generally, Li^+ and Mg^{2+} mass concentration used in this section ranged from 0.164 to 2.456 g L^{-1} , 0.255 to 3.829 g L^{-1} , respectively. These ranges basically are matched with the Li^+ and Mg^{2+} mass concentration from the Hombre Muerto salt lake, Chott Djerid brine, Smackover brine (average), Clayton Valley and Salton Sea (Geothermal Brines), demonstrating common of the selected concentrations used in this section. Moreover, this section also illustrated that the designed SADs polyamide membranes were performance stable at high salt concentrations.

The related explanation has been added in Line 402–403 of the marked version of manuscript.

Table 1 Li^+ and Mg^{2+} mass concentration (g L^{-1}) used in this work when the LiCl and MgCl_2 are 1 g L^{-1} , 2 g L^{-1} , 5 g L^{-1} , 10 g L^{-1} and 15 g L^{-1} , respectively.

	Ion mass concentration (g L^{-1})				
Li^+	0.164	0.328	0.819	1.637	2.456
Mg^{2+}	0.255	0.511	1.276	2.553	3.829

Table 2 Li⁺ and Mg²⁺ mass concentration of various brine of commercial value around the world¹.

Location and feed		Li ⁺ and Mg ²⁺ mass concentration of various brines (g L ⁻¹)		
		Li ⁺	Mg ²⁺	Mg ²⁺ /Li ⁺
Atacama salt lake		1.5–2.42	9.3–13	6.2–5.37
Hombre Muerto salt lake		0.19–0.9	0.18–1.4	0.95–1.56
Uyuni salar brine	Raw1	0.84	16.7	19.88
	Raw2	0.76	17.4	22.89
	After Evaporation	30.3	0.65	0.02
Chott Djerid brine		0.06	3.4	56.67
Smackover brine	Average	0.174	3.465	19.91
	Raw1	0.071	0.28	3.94
Clayton Valley		0.4–0.67	0.53–0.6	0.90–1.33
Qarhan salt lake		0.35	127.9	365.43
X Taijinar salt lake	After Evaporation	6.7	92.43	13.80
	Raw1	0.21	13.5	64.29
Zabuye lake		1.09–1.86	5.28	2.84–4.84
Lungmu Co lake		0.168	14.4	85.71
Yiliping lake		0.259	23.88	92.2
Da Qaidam lake		0.084	9.58	114.1
Dong Taijinar lake		0.16	5.6	35
Chagcam Caka lake		0.28	10.6	37.86
Southern Tibet	Geothermal Brines	0.02–0.239	0.01–0.02	0.08–0.1
Salton Sea	Geothermal Brines	0.1–0.4	0.7–5.7	7–14.25
Cerro Prieto/salt pond	Geothermal Brines	0.393	NA	-
Cesano	Geothermal Brines	0.35	0.012	0.03

Reference

1. Xu, S. et al. Extraction of lithium from Chinese salt-lake brines by membranes: Design and practice, *J. Membr. Sci.* **635**, 119441 (2021).

25. Line 329-330: What are the initial concentrations of Li and Mg in this section? This information is important for understanding the relevance of this study.

Response:

Thank you very much for your suggestion.

Based on your comment, the initial concentrations of Li and Mg has been added in this section. Below are the details.

“Specifically, as demonstrated in Supplementary Table 4, the Mg^{2+}/Li^+ ratio (MLR) of 7.8 represents that mass concentration of Li^+ and Mg^{2+} are 81.87 ppm and 638.19 ppm, respectively. Similarly, MLR is 15.6, having a Li^+ mass concentration of 81.87 ppm, and a Mg^{2+} mass concentration of 1276.38 ppm. And a MLR of 31.2 exhibits mass concentration of Li^+ and Mg^{2+} are 81.87 ppm and 2552.75 ppm, respectively. Moreover, an operational pressure of 1.5 MPa and a MLR of 31.2 were also conducted to investigate the Li^+/Mg^{2+} separation performance.”

Please see Line 412–416 in marked version of manuscript.

26. Line 324: The y-axis of supplementary Fig. 33 is not selectivity, and it is not for Li/Mg; the authors need to update the figure.

Response:

Thank you very much for your suggestion.

Based on your comments, the supplementary Fig. 33 (now Supplementary Fig. 36 in Supplementary Information) on the Li^+/Mg^{2+} selectivity has been corrected and updated.

Below is the corrected figure.

Supplementary Figure 33 | Li⁺/ Mg²⁺ selectivity operational stability of the fabricated polyamide membranes.

27. Line 348: I recommend changing the x-axis of Fig. 4e to a log scale to obtain a better visualization of the data.

Response:

Thank you very much for your suggestion.

According to your recommendation, the x-axis of Fig. 4e has been modified as a log scale for a better visualization of the data. Below is the modified figure.

Fig. 4e Li⁺ rejection as a function of Mg²⁺ rejection.

28. Line 348-350: Consider rephrasing.

Response:

Thank you very much for your suggestion.

Based on your comment, the related sentence has been modified as the follows, "As confirmed in Supplementary Table 4 and Supplementary Fig. 34, since both excellent Mg^{2+} rejection and low Li^+ rejection were obtained in mixed feed solution, and combined the equation 3 in Supplementary Information, the high Li^+/Mg^{2+} selectivity calculated was observed in this work."

Please see Line 447–451 in marked version of manuscript.

29. Line 352-353: What is the rationale for adding the upper bond line? If this is an arbitrary decision, it is recommended to remove it. If is not an arbitrary decision, I recommend providing context.

Response:

Thank you very much for your suggestion.

According to your recommendation, the upper bond line in Fig. 4f has been removed. The related sentence containing the upper bond line also has been deleted.

30. Line 371: What is the rationale for adding an upper bond line? If this is an arbitrary decision, it is recommended to remove it. If is not an arbitrary decision, I recommend providing context.

Response:

Thank you very much for your suggestion.

According to your recommendation, the upper bond line in Fig. 4g has been removed. The related sentence the upper bond line also has been deleted.

31. Line 388-392: Authors need to include a cost analysis to claim that their membrane is cheaper than commercial membranes.

Response:

Thank you very much for your suggestion.

Fig. 1 Module analysis: Calculated Li⁺ recovery (%) vs membrane area (m²) along a simulated module. Feed conditions: Li⁺ concentration of 3.4 mM, Mg²⁺ of 19.6 mM, and operating pressure of 10 bar. Membrane properties: choosing the best-reported data for each type of membrane¹⁻¹².

As shown in Fig. 1, we applied a simulated module to calculate Li recovery vs membrane area. The modelling outcome suggests that, under the same pressure of 10 bar and mixed ion concentration (3.4 mM Li⁺ and 19.6 mM Mg²⁺), a higher Li recovery is achieved with the SADs polyamide membranes in series as compared with other membranes including commercial NF 90 and Desal DL when using the same membrane area. For example, for the A-PA/p-HC-G4D, a 40% Li⁺ recovery in separation process may be achieved with 1.69 m² membrane area, while for the TFN/PEI-3, the used membrane area would be up to 3.48 m². In other words, the larger permeance of the SADs polyamide membranes would allow the use of a smaller membrane area to achieve the same Li⁺ recovery, translating to near proportional savings in capital and membrane replacement costs, also along with a reduction in the power consumption from operation pressure. Finally, we rationally estimated that our designed polyamide membranes would be cheaper than commercial membranes.

The related cost analysis was added in the manuscript, please refer to the

Line 483–493 in marked version of manuscript.

Reference

1. Pramanik, B. K., Asif, M. B., Kentish, S., Nghiem, L. D. & Hai, F. I. Lithium enrichment from a simulated salt lake brine using an integrated nanofiltration-membrane distillation process. *J. Environ. Chem. Eng.* **7**, 103395 (2019).
2. Li, X., Zhang, C., Zhang, S., Li, J., He, B. & Cui, Z. Preparation and characterization of positively charged polyamide composite nanofiltration hollow fiber membrane for lithium and magnesium separation. *Desalination* **369**, 26–36 (2015).
3. Sun, S-Y., Cai, L-J., Nie, X-Y., Song, X. & Yu, J-G. Separation of magnesium and lithium from brine using a Desal nanofiltration membrane. *J. Water Process Eng.* **7**, 210–217 (2015).
4. Yang, Z., Fang, W., Wang, Z., Zhang, R., Zhu, Y. & Jin, J. Dual-skin layer nanofiltration membranes for highly selective $\text{Li}^+/\text{Mg}^{2+}$ separation. *J. Memb. Sci.* **620**, 118862 (2021).
5. He, R., Dong, C., Xu, S., Liu, C., Zhao, S. & He, T. Unprecedented $\text{Mg}^{2+}/\text{Li}^+$ separation using layer-by-layer based nanofiltration hollow fiber membranes. *Desalination* **525**, 115492 (2022).
6. Bi, Q., Zhang, C., Liu, J., Liu, X. & Xu, S. Positively charged zwitterion-carbon nitride functionalized nanofiltration membranes with excellent separation performance of $\text{Mg}^{2+}/\text{Li}^+$ and good antifouling properties. *Sep. Purif. Technol.* **257**, 117959 (2021).
7. Guo, C. et al. Ultra-thin double Janus nanofiltration membrane for separation of Li^+ and Mg^{2+} : “Drag” effect from carboxyl-containing negative interlayer. *Sep. Purif. Technol.* **230**, 115567 (2020).
8. Lu, D. et al. Constructing a selective blocked-nanolayer on nanofiltration membrane via surface-charge inversion for promoting Li^+ permselectivity over Mg^{2+} . *J. Memb. Sci.* **635**, 119504 (2021).
9. Aghili, F., Ghoreyshi, A. A., Van der Bruggen, B. & Rahimpour, A. A highly permeable $\text{UiO-66-NH}_2/\text{polyethyleneimine}$ thin-film nanocomposite membrane for recovery of valuable metal ions from brackish water. *Process Saf. Environ.* **151**, 244–256 (2021).
10. Wang, L. et al. Novel positively charged metal-coordinated nanofiltration membrane for lithium recovery. *ACS Appl. Mater. Inter.* **13**, 16906–16915 (2021).

11. Li, W., Shi, C., Zhou, A., He, X., Sun, Y. & Zhang, J. A positively charged composite nanofiltration membrane modified by EDTA for LiCl/MgCl₂ separation. *Sep. Purif. Technol.* **186**, 233–242 (2017).
12. He, R., Xu, S., Wang, R., Bai, B., Lin, S. & He, T. Polyelectrolyte-based nanofiltration membranes with exceptional performance in Mg²⁺/Li⁺ separation in a wide range of solution conditions. *J. Memb. Sci.* **663**, 121027 (2022).

32. Line 404-405: Phrase does not seem connected to the rest of the paragraph. Consider rephrasing.

Response:

Thank you very much for your suggestion.

The inappropriate phrase “And adsorption behaviors of NaCl using QCM.” has been deleted. The phrase to describe the adsorption behaviors of NaCl for the nanofilms was added in the Line 542–546 in marked version of manuscript.

33. Line 408-410: I do not follow the discussion in this sentence. Are the authors discussing the importance of the size, uniformity, or both? Therefore, rephrasing is recommended.

Response:

Thank you very much for your suggestion.

Based on your comment, the related sentence has been modified as follows:

“It is known that separation mechanism for nanofiltration membrane mainly includes size sieving and Donnan effect. Therefore, efficient pore size range and uniform pore structure are significant for polyamide nanofilm in terms of ion separation. As demonstrated in Fig. 5a, PA networks incorporating SADs (PA/DA-G4D, PA/BA-G4D and PA/p-HC-G4D) have no pore structure greater than 5 Å as relative to the pristine PA polymer, exhibiting a narrower pore size range and a more uniform pore structure. These pore structure features are more conducive to conduct efficient ion separation.”

Please refer to the Line 509–516 in marked version of manuscript.

34. Line 410-417: It is difficult for me to understand how a material can be

simultaneously more porous and denser. Seems to me that these two features are inversely proportional.

Response:

Thank you very much for your suggestion.

As you commented, for a material, it is true that it cannot simultaneously be more porous and denser. However, in this work, there were actually two materials in polyamide nanofilm. One was the pristine polyamide polymer, which were consisted of TMC and PIP, possessing a semi-aromatic structure. Another was the dendrimer with intrinsic nanovoids, showing a full-aromatic structure. It is known that density also is related with the chemical structure^{1,2}. It has been confirmed that the semi-aromatic polyamide polymer has a lower density than that of the full-aromatic polyamide polymer¹. For example, density of the TMC-PIP (commercial NF 270) is $0.987 \pm 0.26 \text{ g cm}^{-3}$, whereas that of the TMC-MPD (the average active layer density of commercial membranes) is $1.267 \pm 0.21 \text{ g cm}^{-3}$ ¹. Since the incorporation of dendrimer with full-aromatic structure, it increased the density of the polyamide polymer, as demonstrated in Fig. 5d in Manuscript. Moreover, due to the intrinsic nanovoids of the designed dendrimers, it also enhanced the porosity of the resultant polyamide nanofilm, as demonstrated in Fig. 5c in Manuscript. Hence, the feature of simultaneously more porous and denser for the polyamide polymer nanofilm was achieved by the design of chemical structure of polyamide.

The related explanation has been added in the marked version of manuscript, please refer to the Line 516–523.

Reference

1. Lin, L., Feng, C., Lopez, R. & Coronell, O. Identifying facile and accurate methods to measure the thickness of the active layers of thin-film composite membranes—a comparison of seven characterization techniques. *J. Membr. Sci.* **498**, 167–179 (2016).
2. Zhang, X., Cahill, D. G., Coronell, O. & Marinas, B. J. Absorption of water in the active layer of reverse osmosis membranes. *J. Membr. Sci.* **331**, 143–151 (2009).

35. Line 435: What is the difference in the NaCl adsorption?

Response:

Thank you very much for your suggestion.

As shown in Fig. 5e, adsorption behaviors of Cl^- (NaCl) on the pristine A-PA nanofilm and the SADs PA nanofilms as a function of time exhibit a significant difference. Since a uniform distribution of ultrasmall pore was observed for the PA containing SADs polymer compared to the pristine PA, it showed a less NaCl adsorption content. This difference on the NaCl adsorption of pristine A-PA nanofilm and the SADs PA nanofilms further confirmed that the link between chemistry structure and ion separation performance.

The related explanation has been added in the marked version of manuscript, please refer to the Line 542–546.

Response to Reviewer 3

Reviewer #3 (Remarks to the Author):

This paper deals with precision separation of mono/di-valent ions. Specifically Li⁺/Mg²⁺ separation (and associated Cl⁻/SO₄²⁻) separation. The topic is important as resource recovery (as exemplified in the concept of 'brine mining' is attracting more and more attention - and seen as an important aspect of achieving circularity in resource management.

The paper describes synthesis, characterization and functional tests of membranes based on dendrimers with carboxyl and phenolic hydroxyl groups embedded in a polyamide structure.

Overall the paper builds on a sound methodology. The experimental results are generally well presented and impressive Li⁺/Mg²⁺ selectivities (from the various dendrimer-based embodiments) are achieved. This is indeed a promising result in the development of membranes for Li⁺-recovery.

There are however a few issues which needs to be addressed.

Response:

Thanks very much for your kind comments and evaluation. We have carefully read your comments, gave some explanations and addressed them one-by-one.

1. First, the use of dendrimer-based membranes for ion separation is not new - and citations to recent papers (e.g. Z-L, Qiu et al. ACS Nano 15, 7522-7535, 2021) are missing.

Response:

Thank you very much for your suggestion.

Based on your comment, the recent papers (e.g. Z-L, Qiu et al. ACS Nano 15, 7522–7535, 2021) about the use of dendrimer-based membranes has been added in the revised version of manuscript. Please see Line 73 for the reference of "ionic dendrimer".

2. Second, for the modeling work: why was the B3LYP functional chosen? There are other DFT functionals each with advantages and disadvantages (for a nice review see N. Mardirossian & M. Head-Gordon. MOLECULAR PHYSICS, 2017,

115(19), 2315-2372).

Response:

Thank you very much for your suggestion.

Based on your comment, the following is the reason that we chose the B3LYP functional.

Since the charge information in the molecular topology established online through ATB may not be accurate, in this case, we have to quantitatively calculate the molecular charge and correct the charge part in the molecular topology information output by ATB.

Consequently, applying the B3LYP basis set to conduct the correction of charge information is because that, the accuracy of this basis set is sufficient for calculating atomic charge, and the time consumption of such basis set is short, which can obtain the required atomic charge information faster.

Besides, the related reference (Mardirossian, N. & Head-Gordon, M. MOLECULAR PHYSICS, 115 (19), 2017 (2315–2372)) also was added in the manuscript. Please refer to Line 506 (reference 53) in the revised version of manuscript.

3. Third, what role (if any) does electronic polarization of the constituent molecular structure play in the separation mechanism and how is this addressed in the MD simulations?

Response:

Thank you very much for your suggestion.

We rationally estimated that there is no electronic polarization in the molecular structure of nanofilm. Below are the detail explanations.

Firstly, since ensuring a net charge of 0 in the simulation box is needed for MD simulation, we have calculated and balanced all charges of the functional groups that need to be protonated throughout the entire building process for nanofilm.

Secondly, according to separation mechanism of nanofiltration membrane, we rationally consider that membrane structure has a significant influence on membrane separation performance. Therefore, during MD simulation studies, the effect of dendrimer incorporation on pore structure feature of nanofilm was primarily investigated.

On the other hand, if electronic polarization occurs, it would be equivalent

to quite severe conditions in the experiment, or an equal number of opposite charges are not added to balance the charge in the simulation box, which will not reflect the requirements of the practical experiment or simulation.

It is known that nanofiltration membrane is driven by pressure and does not need an external electric field. Therefore, generally speaking, electronic polarization is not occurred in the molecular structure. Taken together, there is no existence of electronic polarization, and it will not affect the separation mechanism.

4. Finally, how does the MD PBC setup (which is indeed a good starting point) reflect the true corrugated membrane surface - and what implications does this simplification have for the results presented?

Response:

Thank you very much for your suggestion.

During the process of building the nanofilm model using MD simulations, we need to set PBCs in the three directions of XYZ to prevent molecules from running out of the simulation box constructed. Meantime, this PBC setup also ensures occurrence of the follow situation, where the lack of new added molecules in the opposite direction leads to a decrease in the number of molecules in the simulation box, ultimately affecting the construction of the model.

If the performance of the nanofilm simulated needs to be tested, periodicity in the directions X and Y of the membrane is required to be setted, while aperiodicity treatment is carried out in the Z direction, then producing rough surfaces that is similar to the real membrane surface. Since this work only explores the effects of different dendrimer doping on the three-dimensional structure of the nanofilm, it does not involve the surface properties of the membrane, but only the internal structure of the membrane. Therefore, the model constructed was not subjected to periodic processing. This simplification also had no impact on the presented results.

REVIEWERS' COMMENTS

Reviewer #1 (Remarks to the Author):

the authors have addressed my concerns.

Reviewer #2 (Remarks to the Author):

The authors addressed most of the comments from my initial review. However, there are two comments that were partially addressed.

Comment 27: The authors suggested that the graph was updated, but it remained the same in the latest version.

Comment 31: The authors partially addressed my comment about cost. However the line "translating to near proportional savings in capital and membrane replacement costs" could be misleading. To claim that their membrane will save capital and replacement costs they need to add the cost of manufacturing their membrane and compare this value with the cost of manufacturing a commercial membrane. In other words, if replacing a commercial membranes costs \$10 and replacement their membrane costs \$1000, the commercial membrane would be cheaper even if it has to be replaced more frequently.

Reviewer #3 (Remarks to the Author):

All my queries/comments have been addressed - thank you!

Reviewer #2 (Remarks to the Author):

The authors addressed most of the comments from my initial review. However, there are two comments that were partially addressed.

Comment 27: The authors suggested that the graph was updated, but it remained the same in the latest version.

Comment 31: The authors partially addressed my comment about cost. However the line "translating to near proportional savings in capital and membrane replacement costs" could be misleading. To claim that their membrane will save capital and replacement costs they need to add the cost of manufacturing their membrane and compare this value with the cost of manufacturing a commercial membrane. In other words, if replacing a commercial membranes costs \$10 and replacement their membrane costs \$1000, the commercial membrane would be cheaper even if it has to be replaced more frequently.

Response to Reviewer 2

Comment 27. The authors suggested that the graph was updated, but it remained the same in the latest version.

Response:

Thank you very much for your suggestion.

The x-axis of Fig. 4e in manuscript has been modified as a log scale for a better visualization of the data. The following Fig. 4e is the updated graph. Such updated graph was also added in the latest manuscript.

Fig. 4e Li^+ rejection as a function of Mg^{2+} rejection.

Comment 31. The authors partially addressed my comment about cost. However the line "translating to near proportional savings in capital and membrane replacement costs" could be misleading. To claim that their membrane will save capital and replacement costs they need to add the cost of manufacturing their membrane and compare this value with the cost of manufacturing a commercial membrane. In other words, if replacing a commercial membranes costs \$10 and replacement their membrane costs \$1000, the commercial membrane would be cheaper even if it has to be replaced more frequently.

Response:

Thank you very much for your suggestion.

Your view of point on cost is equivalent to be true. When we talk about the advantages of a new type of membrane in terms of saving capital and replacement costs, it means that the challenging issues that need to be dissolved are changed from basic research, scaling up production to commercial application. The cost of manufacturing such membrane need to be evaluated.

Based on our knowledge of membrane production costs, non-woven fabrics and polysulfone ultrafiltration membranes account for 70% to 80% of the cost of nanofiltration membranes, aqueous amine solution and organic phase chloride solution account for 10% to 15% of the cost of nanofiltration membranes. The dendrimer content used in this work accounts for 5% to 10% of the amine content.

According to the high productivity of the DA-G4D, BA-G4D and p-HC-G4D

dendrimers and their cheap synthesis raw materials, we assume that the costs to synthesize these dendrimers would be decreased with the scale up. In this case, we briefly suppose the costs of these dendrimers used are twice or triple the cost of the amine and chloride used. After appropriated calculated, we estimate that, due to the application of the same interfacial polymerization process and with the scaling up production for these self-assembled dendrimer polyamide membranes, the cost of manufacturing is 1.2 to 1.3 times that of commercial membranes. Hence, considering the saving in membrane area used, we estimate that the resulting membranes also exhibit a potential advantage in terms of membrane replacement costs.

Based on the above discussion, to avoid possible misleading, the discussion on the cost analysis has been corrected as follows:

“In other words, the larger permeance of the SADs polyamide membranes would allow the use of a smaller membrane area to achieve the same Li⁺ recovery. In this case, a reduction in the power consumption from operation pressure would be achieved¹⁶. Moreover, with the scaling up production for these SADs polyamide membranes and the costs that synthesize these dendrimers would be decreased accordingly, a potential advantage in terms of membrane replacement costs can also be expected.”